# BAYESIAN PRIMITIVE DISTRIBUTING FOR COMPOSITIONAL ZERO-SHOT LEARNING

## ABSTRACT

Compositional zero-shot learning (CZSL) aims to recognize unseen attribute-object combinations by learning primitive concepts (*i.e.*, attribute and object) from seen compositions. Existing CZSL solutions typically harness the power of vision-language models like CLIP via textual prompt tuning and visual adapters. However, they independently learn one deterministic textual prompt for each primitive or compositional labels, ignoring both the inherent semantic diversity within each primitive and the semantic relationships between primitive concepts and their compositions. In this paper, we propose BAYECZSL, a novel Bayesian-induced framework that learns probability distributions over each primitive textual prompt from a Bayesian perspective. Specifically, BAYECZSL models image-specific primitive textual prompts as learnable probability distributions to capture intra-primitive diversity. Building on these primitive distributions, we aggregate learned probability distributions from attribute and object branches to form compositional prompt space via Compositional Distribution Synthesis strategy, thus capturing the semantic relationships between primitive concepts and their compositions. Moreover, Three-path Distribution Enhancement module is introduced to transform initial distributions into expressive ones via invertible mappings. Finally, these enhanced distributions are sampled to generate diverse textual prompts, achieving more comprehensive coverage of the prompt space and generalizing to unseen compositions. Extensive experiments on multiple CZSL benchmarks demonstrate the superiority of our BAYECZSL. Code will be released.

## 1 INTRODUCTION

Humans possess the remarkable capacity to effortlessly recombine previously encountered attributes and objects [27], enabling them to reason over unseen compositional concepts [45, 2]. For instance, even without having seen a wildebeest, one can readily imagine its appearance by integrating the notions of "horns" and "horse". Endowing machines with such compositional reasoning capabilities [26] is the goal of Compositional Zero-Shot Learning (CZSL) [14, 16], which aims to recognize novel attribute-object compositions by leveraging knowledge from previously seen compositions.

Traditional CZSL solutions [14, 24, 32, 58, 46, 13, 62] typically focus on compositional learning via aligning visual features extracted from a pre-trained vision encoder and textual embed-

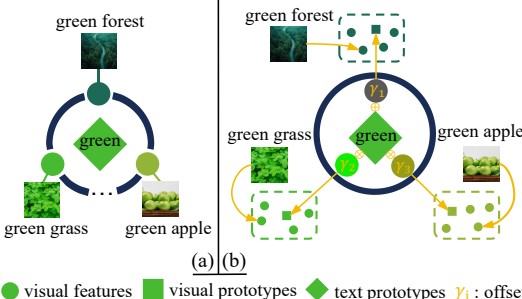

Figure 1: (a) Existing CLIP-based methods only rely on a single textual prompt to represent each primitive concept, ignoring the intra-primitive diversity when involved in different compositions. (b) Our method learns a probability distribution over each primitive prompt to model the intra-primitive variance.

dings of attribute-object labels. Due to the availability of pretrained vision-language models (*e.g.*, CLIP), recent approaches [30, 8, 20, 18, 66, 49] harness the powerful visual-semantic aligning capabilities of CLIP for recognizing attribute-object compositions via various finetuning strategies, *e.g.*, prompt tuning [28, 34], adapters [68, 9], and cross-attention mechanism [11, 23]. Though these methods show impressive performance, they exhibit two key limitations: **First**, they typically

learn one single deterministic textual prompt for each primitive concept (attribute or object) and their compositions, which is oversimplified and struggles to capture the complex inherent semantic diversity within each primitive [7, 35]. For example, the attribute "old" conveys distinct semantic meanings when applied in different compositions, *e.g.*, "old dog" and "old town". Thus, we argue that one single learnable textual prompt is insufficient to capture intra-primitive variation, and learning probability distributions over textual prompts to expand the prompt space is necessary to model the natural diversities of primitives (Fig. 1). **Second**, they treat attribute, object, and compositional prompts independently, ignoring the rich relational structure between primitives and their compositions. As a result, the learned prompts remain overly isolated and fail to exploit cross-branch synergies, where different branches of the model complement and enhance each other's contributions, leading to limited generalization capacity when encountering novel combinations.

To address these limitations, we present BAYECZSL, a Bayesian-induced framework for CZSL that explicitly models probability distributions over each primitive textual prompt (*i.e.*, attribute and object) from a Bayesian inference perspective. *Different* from representing each attribute or object with one single prompt, BAYECZSL learns distributional prompts that capture the natural variability of primitives. Building on learned probability distributions from primitive branches, we aggregate such two distributions to form a compositional prompt space via *Compositional Distribution Synthesis*, explicitly capturing the semantic relationships between primitive concepts and compositions.

Concretely, BAYECZSL starts by learning the probability distributions for each primitive textual prompt via **Bayesian-induced Primitive Distribution learning**, which effectively represent the intra-primitive diversity and reduce overfitting on seen attribute–object combinations. To introduce rich visual semantics into the text prompt space, our constructed primitive prompt distributions are dynamically adapting based on the primitive-wise visual features. Beyond modeling the attribute and object distributions separately, we further employ the **Compositional Distribution Synthesis** module, which aggregates the learned probability distribution of both attribute and object branches into a unified compositional prompt space, thus capturing the rich semantic relationships between primitive concepts and their compositions. Moreover, to better approximate complex prompt distributions for intra-primitive modeling and unseen composition generalization, we adopt a **Three-path Distribution Enhancement** module, which transforms simple initial primitive and composition distributions into more flexible and expressive ones via a sequence of invertible mappings. Finally, we draw multiple Monte-Carlo samples from these enhanced distributions and mix them with original textual prompts to enhance the coverage of the textual prompt space, thus reducing overfitting on seen attribute–object combinations and improving generalization on unseen compositions.

The contributions of this work can be summarized as follows: **First**, we revisit CZSL task from the Bayesian inference view, and learn probability distributions over attribute and object prompts to explicitly model intra-primitive variability and semantic uncertainty. By distribution sampling and distribution regularization of the textual prompt space, BAYECZSL reduces overfitting to seen compositions, and improve generalization on unseen compositions. **Second**, we introduce a novel Compositional Distribution Synthesis mechanism that aggregates the probability distribution of attribute and object branches to form the compositional prompt space, thus capturing the rich semantic relationships between primitive concepts and their compositions. **Third**, we develop a Three-path Distribution Enhancement module to transform base prompt distributions into expressive ones, thus facilitating diverse prompt sampling for comprehensive intra-primitive modeling.

Extensive experiments on three challenging benchmark datasets (MIT-States [17], UT-Zappos [65], and C-GQA [42]) demonstrates BAYECZSL outperforms existing CZSL methods by a large margin in both *Close-World (CW)* and *Open-World (OW)* settings (§4.2). Concretely, on the *CW* setting, BAYECZSL exceeds the current state-of-the-art methods by up to **+8.9%** and **+3.2%** relative AUC improvement on UT-Zappos and C-GQA. Under the more challenging *OW* setting, BAYECZSL still surpasses the best CLIP-based method by up to **+7.0%** and **+14.8%** relative AUC improvement on UT-Zappos and C-GQA. In §4.3, we further conduct extensive ablation studies to validate the effectiveness of each model component.

## 2 RELATED WORK

**Compositional Zero-shot Learning (CZSL).** The objective of CZSL is to recognize unseen attribute-object compositions by learning a comprehensive knowledge of seen compositions. Early CZSL

approaches generally follow two main strategies. The first strategy extracts composed attribute-object semantic features through a transformation function and performs recognition directly with a classifier [33, 41, 43, 48, 59, 23]. The second strategy employs a disentangler to separate the original image features into distinct attribute and object representations, which are then independently classified using two separate classifiers [54, 67, 52, 11, 64, 31, 63]. However, all of these methods require learning the alignment between image features and text embeddings from scratch, which makes them prone to overfitting on the seen compositions. Recent studies have increasingly focused on utilizing pre-trained vision-language models (VLMs) to tackle the challenge of compositional zero-shot learning. Troika [15] proposes a multi-path paradigm to jointly model the attribute, object, and composition. DFSP [36] proposes a cross-modal decomposed fusion module that leverages a disentangler and constructs a vector combination of learnable soft prompts with attribute and object to capture more detailed features. PLID [4] integrates pretrained large language models to construct diverse and expressive prompt distributions, orthogonal to prior work on soft, hard, and distributional prompting. In contrast to previous methods, our BAYECZSL models the distribution of textual prompts and leverages sampling to explore the prompt space based on a multi-path paradigm, thereby enhancing performance in compositional zero-shot learning (CZSL).

**Prompt Learning in VLMs.** As an efficient adaptation strategy, prompt learning enables Vision-Language Models (VLMs) to be customized for specific tasks. Vision-Language Models (VLMs) such as the CLIP are pre-trained on large-scale image-text pairs, recently have demonstrated their effectiveness in diverse vision-language applications, most notably in zero-shot recognition [71, 57, 55, 19]. In early prompting methods, like the hard prompt used in CLIP, heuristic templates such as *"a photo of* [CLS]" are used as textual inputs. Recently, the methods in CoOp [67], CoCoOp [69] and CSP [44] use soft prompt tuning. The former treats the context of class names as learnable prompt tokens, while the latter uses a fixed template for the context and treats the class names themselves as learnable prompt tokens. In CDS-CZSL [34], the entire prompt is further treated as learnable parameters, enabling the model to capture task-relevant information more precisely. However, the prompts used by these approaches are fixed and insufficiently diverse to represent the wide appearance variations in fine-grained visual data, making them susceptible to overfitting on the training set. To address this problem, ProDA [37] explicitly employs a set of soft prompts to build class-specific gaussian distributions, leading to improved zero-shot performance. PLO [29] further promotes finer-grained understanding by progressively and adaptively observing primitives, employing a staged observation approach to prevent model overfitting. Recent work [3, 52, 56, 5] assumes that the latent embedding of the prompt input follows a gaussian prior and utilizes variational inference to learn the latent distribution. In this paper, we introduce a Bayesian-induced framework that represents textual prompts as probability distributions, which encourages diverse prompt generation and strengthens generalization to unseen compositions. This probabilistic modeling facilitates broader coverage of the prompt space and captures richer semantic relationships.

# 3 METHODOLOGY

## 3.1 PROBLEM STATEMENT

Given the attribute set $\mathcal{A} = \{a_1, a_2, \ldots, a_{|\mathcal{A}|}\}$ and the object set $\mathcal{O} = \{o_1, o_2, \ldots, o_{|\mathcal{O}|}\}$ as primitive concepts, the compositional space $\mathcal{C}$ is defined as their Cartesian product: $\mathcal{C} = \mathcal{A} \times \mathcal{O}$. The objective of the CZSL task is to recognize images belonging to a compositional category $y \in \mathcal{C}$, where the compositional space $\mathcal{C}$ is subsequently partitioned into two disjoint subsets: the seen composition set $\mathcal{C}_s$ and the unseen composition set $\mathcal{C}_u$, such that $\mathcal{C}_s \cap \mathcal{C}_u = \varnothing$. The training set is defined as $\mathcal{T} = \{(x_k, c_k) \mid x_k \in \mathcal{X}, c_k \in \mathcal{C}_s\}$, where $\mathcal{X}$ represents the image space. In the *Closed-World (CW)* setting, the target set is defined as $\mathcal{C}_t = \mathcal{C}_s \cup \mathcal{C}_u$, where only compositions of the known space are considered. In contrast, the *Open-World (OW)* setting assumes that the target set consists of all possible permutations of attribute-object pairs, *i.e.,* $\mathcal{C}_t = \mathcal{C}$.

## 3.2 BASELINE ARCHITECTURE

**Textual Prompt Extraction.** Our framework is built upon a three-path paradigm [15, 36], which jointly recognizes three types of semantic components: attributes, objects, and attribute-object compositions. Following prior work in CZSL [15], we construct prompt representations using a soft and learnable prompting strategy for the three aforementioned semantic components. For a

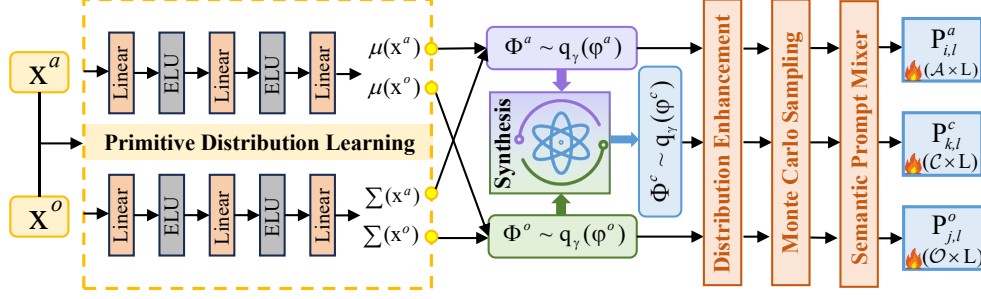

**(a)Training and Inference**

**(b) Bayesian-induced learning**

☐ Image Features  ☐ Text Features  🔥 Learnable Parameters  $\mu(x)$ Distributional Mean  $\sum(x)$ Distributional Covariance

Figure 2: The overview of BAYECZSL: (a) Training and inference (§3.2 and §3.4); (b) Bayesian-induced learning (§3.3): Bayesian-induced Primitive Distribution Learning, Compositional Distribution Synthesis, and Three-path Distribution Enhancement.

given attribute-object composition $c_{i,j} = \langle a_i, o_j \rangle$, we create prompts using a three-path paradigm, *i.e.,* attribute prompt $\mathbf{P}_i^a = [p_{i,1}^a, \ldots, p_{i,m}^a, \mathbf{v}_i^a]$, object prompt $\mathbf{P}_j^o = [p_{j,1}^o, \ldots, p_{j,m}^o, \mathbf{v}_j^o]$ and composition prompt $\mathbf{P}_k^c = [p_{k,1}^c, \ldots, p_{k,m}^c, \mathbf{v}_{k,a}^c, \mathbf{v}_{k,o}^c]$. We initialize the learnable prompt prefixes $[p_{i,1}^a, \ldots, p_{i,m}^a], [p_{j,1}^o, \ldots, p_{j,m}^o]$ and $[p_{k,1}^c, \ldots, p_{k,m}^c]$ with the phrase *"a photo of"*, serving as a semantic prior to guide prompt optimization. In addition, $\mathbf{v}_i^a$, $\mathbf{v}_j^o$ and $\mathbf{v}_k^c$ denote the trainable vocabulary embeddings corresponding to the attribute $a_i$, object $o_j$ and composition $c_k$. These fully trainable prompts are subsequently passed through the text encoder $E_{\text{txt}}$ to derive the prompt representations for each branch, formulated as follows:

$$\mathbf{t}_i^a = E_{\text{txt}}(\mathbf{P}_i^a), \quad \mathbf{t}_j^o = E_{\text{txt}}(\mathbf{P}_j^o), \quad \mathbf{t}_k^c = E_{\text{txt}}(\mathbf{P}_k^c). \tag{1}$$

**Visual Feature Extraction.** Following the prior works [23, 15], we incorporate adapter modules [34, 68] to adapt the image encoder while keeping its original parameters frozen. Given an input image $x \in \mathbb{R}^{H \times W \times 3}$, the visual encoder $E_{\text{img}}$ of CLIP [50] is employed to obtain the image representation $\mathbf{x} \in \mathbb{R}^D$. We treat the image representation $\mathbf{x}$ as compositional features $\mathbf{x}^c$, and employ an attribute disentangler $D_a$ and an object disentangler $D_o$ to decouple the composition feature $\mathbf{x}^c$ into the attribute feature $\mathbf{x}^a$ and an object feature $\mathbf{x}^o$ as:

$$\mathbf{x}^c = \mathbf{x}, \quad \mathbf{x}^a = D_a(\mathbf{x}^c), \quad \mathbf{x}^o = D_o(\mathbf{x}^c), \tag{2}$$

where $D_a$ and $D_o$ are implemented as two separate MLPs [53].

**Three-path paradigm Training.** Based on the three-path paradigm, we provide each branch with corresponding prompts and visual representations, and separately compute the probabilities of assigning attribute $a_i$, object $o_j$, and composition $c_k$ labels to the image. To enable the recognition of primitive concepts and their compositions within each branch, we also utilize three separate cross-entropy loss functions. They are expressed as follows:

$$p(a_i \mid x) = \frac{\exp(\mathbf{x}^a \cdot \mathbf{t}_i^a / \tau)}{\sum_{n=1}^{|A|} \exp(\mathbf{x}^a \cdot \mathbf{t}_n^a / \tau)}, \quad \mathcal{L}_a = \frac{1}{|\mathcal{X}|} \sum_{x \in \mathcal{X}} -\log p(a \mid x), \tag{3}$$

$$p(o_j \mid x) = \frac{\exp(\mathbf{x}^o \cdot \mathbf{t}_j^o / \tau)}{\sum_{n=1}^{|O|} \exp(\mathbf{x}^o \cdot \mathbf{t}_n^o / \tau)}, \quad \mathcal{L}_o = \frac{1}{|\mathcal{X}|} \sum_{x \in \mathcal{X}} -\log p(o \mid x), \tag{4}$$

$$p(c_k \mid x) = \frac{\exp(\mathbf{x}^c \cdot \mathbf{t}_k^c / \tau)}{\sum_{n=1}^{|C_t|} \exp(\mathbf{x}^c \cdot \mathbf{t}_n^c / \tau)}, \quad \mathcal{L}_c = \frac{1}{|\mathcal{X}|} \sum_{x \in \mathcal{X}} -\log p(c \mid x), \tag{5}$$

where $\tau \in \mathbb{R}$ is the pre-defined temperature parameter in CLIP. Thus, the three-path classification loss can be formulated as:

$$\mathcal{L}_{\text{aoc}} = \beta_a \mathcal{L}_a + \beta_o \mathcal{L}_o + \beta_c \mathcal{L}_c, \tag{6}$$

where $\beta_a, \beta_o, \beta_c$ are all set to 1, following [15]. More ablation studies can be found in Appendix §G.

**Motivation.** Though impressive, these methods learn one single-form textual prompt for each primitive and compositional label, ignoring the semantic diversity of primitive concepts across different compositions and struggling to generalize to unseen compositions. Moreover, they overlook the rich semantic relationships between primitive concepts and their compositions. To address these limitations, we propose BAYECZSL in Fig. 2, which models primitive textual prompts as probability distributions from a Bayesian inference perspective, capturing both intra-primitive diversity and inter-primitive relationships. In particular, our model first learns image-specific probabilistic distributions over each primitive concept via **Bayesian-induced Primitive Distribution learning**. Then our proposed **Compositional Distribution Synthesis** module aggregates the attribute and object probability distribution to form the compositional prompt space. Monte Carlo sampling [40] is applied to these distributions, and the sampled results are fused with the original textual prompts to enhance the coverage of the textual prompt space. Moreover we employ **Three-path Distribution Enhancement** module to better estimating the textual prompt distribution.

### 3.3 BAYESIAN-INDUCED LEARNING

**Bayesian-induced Primitive Distribution Learning.** We define the training set $\mathcal{T} = \{(x_k, c_k) \mid x_k \in \mathcal{X}, c_k \in \mathcal{C}_s\}$, where $x_k$ denotes the input image and $c_k$ represents the associated compositional label. For the attribute and object branches, we construct primitive prompt distributions by embedding the contextual information of each primitive as a D-dimensional random vector $\Phi$, leading to the following posterior distribution:

$$p(\Phi \mid \mathcal{T}) = \frac{p(\mathcal{T} \mid \Phi)\, p(\Phi)}{p(\mathcal{T})}. \tag{7}$$

Because calculating $p(\mathcal{T})$ is tractable, we adopt variational inference [21] with a parameterized distribution $q_\gamma(\Phi)$ to approximate the posterior. This approximation enables the estimation of the marginal likelihood $p(\mathcal{T})$. It is necessary to minimize the KL divergence between $q_\gamma(\Phi)$ and $p(\Phi)$:

$$D_{\text{KL}}(q_\gamma(\Phi) \,\|\, p(\Phi)) = \int q_\gamma(\Phi) \log \frac{q_\gamma(\Phi)}{p(\Phi)}\, d\Phi. \tag{8}$$

Using Jensen's inequality, we derive a variational lower bound on the logarithmic marginal likelihood of the training data:

$$\log p(\mathcal{T}) = \log \int p(\mathcal{T} \mid \Phi) p(\Phi)\, d\Phi \tag{9}$$

$$\geq \mathbb{E}_{q_\gamma(\Phi|\mathcal{T})}\big[\log p(\mathcal{T}, \Phi) - \log q_\gamma(\Phi \mid \mathcal{T})\big] \tag{10}$$

$$= \mathbb{E}_{q_\gamma(\Phi|\mathcal{T})}\big[\log p(\mathcal{T} \mid \Phi)\big] - D_{\text{KL}}(q_\gamma(\Phi \mid \mathcal{T}) \,\|\, p(\Phi)) = -\mathcal{L}_{\text{pri}}(\mathcal{T}). \tag{11}$$

We obtain the primitive variational distribution $q_\gamma(\Phi)$ by minimizing the loss function $\mathcal{L}_{pri}$, which corresponds to the negative evidence lower bound (ELBO).

Given the attribute and object visual features $\mathbf{x}^a$ and $\mathbf{x}^o$, we employ Bayesian inference to map each primitive feature into a distributional representation. Following standard variational optimization practices [10, 25], we model the residual distributions of the two primitives as gaussian, with $\mu(\mathbf{x}^a) \in \mathbb{R}^C, \Sigma(\mathbf{x}^a) \in \mathbb{R}^C$ and $\mu(\mathbf{x}^o) \in \mathbb{R}^C, \Sigma(\mathbf{x}^o) \in \mathbb{R}^C$ estimated from the image features via a three-layer network with ELU activations [6]. Finally, we obtain the variational distributions $\Phi^a \sim q_\gamma(\Phi^a)$ and $\Phi^o \sim q_\gamma(\Phi^o)$ for these two primitives, which capture the semantic uncertainty of each primitive in the prompt space.

**Compositional Distribution Synthesis.** To capture the rich semantic relationships between primitive concepts and their compositions, we adopt a variance-inverse weighted Gaussian fusion strategy, which combines the attribute and object distributions to inform the compositional distributions. Specifically, given the primitive distributions $\Phi^a \sim \mathcal{N}(\mu(\mathbf{x}^a), \Sigma(\mathbf{x}^a))$ and $\Phi^o \sim \mathcal{N}(\mu(\mathbf{x}^o), \Sigma(\mathbf{x}^o))$, we fuse the two primitive distributions as follows:

$$\Sigma(\mathbf{x}^c) = \big[\Sigma(\mathbf{x}^a)^{-1} + \Sigma(\mathbf{x}^o)^{-1}\big]^{-1}, \quad \mu(\mathbf{x}^c) = \Sigma(\mathbf{x}^c)\big[\Sigma(\mathbf{x}^a)^{-1}\mu(\mathbf{x}^a) + \Sigma(\mathbf{x}^o)^{-1}\mu(\mathbf{x}^o)\big]. \tag{12}$$

As such, we obtain the compositional prompt distribution $\Phi_c \sim \mathcal{N}(\mu(\mathbf{x}^c), \Sigma(\mathbf{x}^c))$. This compositional prompt distribution serves as an auxiliary prior, encouraging the model to attend to primitive-relevant regions and to better generalize to unseen attribute–object pairs.

**Three-path Distribution Enhancement.** Employing more expressive posterior approximations enhances the ability to approximate the prompt distribution with greater fidelity, capturing its inherent uncertainty and structural complexity more effectively [51]. Accordingly, a distribution enhancement module is designed, in which a simple probability distribution $q_0(\Phi_0)$ is transformed into a more complex distribution $q_N(\Phi_N)$ through a sequence of invertible mappings $f_N$:

$$\Phi_N = f_N(f_{N-1}(\cdots f_1(\Phi_0))). \tag{13}$$

To improve computational efficiency, a linear-time transformation is employed, defined as:

$$f(\Phi) = \Phi + v \operatorname{Tanh}\left(w^\top \Phi + b\right), \tag{14}$$

where the parameters $w, v \in \mathbb{R}^C$ and $b \in \mathbb{R}$ are trainable, and $\operatorname{Tanh}(\cdot)$ denotes the Tanh activation function [53]. The new distribution after $N$ transformations is expressed as:

$$\log q_N(\Phi_N) = \log q_0(\Phi_0) - \sum_{n=1}^{N} \log \left|1 + v_n^\top y'\left(w^\top \Phi_n + b\right)w\right|. \tag{15}$$

By substituting $q_\gamma(\Phi \mid \mathcal{T})$ in Eq. 11 with the transformed distribution $q_N(\Phi_N)$, the objective function for optimizing the Bayesian-induced framework can be formulated as:

$$\begin{aligned}
\mathcal{L}_\mathrm{p}(\mathcal{T}) &= \mathbb{E}_{q_\gamma(\Phi|\mathcal{T})}\left[\log p(\mathcal{T} \mid \Phi)\right] - D_{\mathrm{KL}}(q_\gamma(\Phi \mid \mathcal{T}) \,\|\, p(\Phi)) \\
&= \mathbb{E}_{q_0(\Phi_0)}\left[\log q_0(\Phi_0) - \sum_{n=1}^{N} \log \left|1 + v_n^\top y'\left(w^\top \Phi_n + b\right)w\right|\right] \\
&\quad - \mathbb{E}_{q_0(\Phi_0)}\left[\log p(\Phi_N)\right] - \mathbb{E}_{q_0(\Phi_0)}\left[\log p(\mathcal{T} \mid \Phi_N)\right].
\end{aligned} \tag{16}$$

We assume that the prior follows a standard normal distribution, *i.e.*, $p = \mathcal{N}(0, \mathbf{I})$. The initial density $q_0$ is modeled as a multivariate normal distribution, specifically, $q_0 = \mathcal{N}(\mu(\mathbf{x}), \Sigma(\mathbf{x}))$, where the mean $\mu \in \mathbb{R}^C$ and the diagonal covariance matrix $\operatorname{diag}(\Sigma) = \sigma \in \mathbb{R}^C$ are conditioned on the input vector $\mathbf{x} \in \mathbb{R}^C$. Both $\mu$ and $\Sigma$ are parameterized by three consecutive linear layers.

Based on different input $\mathbf{x}$ provided to the distribution enhancement module, we obtain the image-conditioned distributions: the attribute distribution $q_N(\Phi_N^a)$, the object distribution $q_N(\Phi_N^o)$, and the auxiliary compositional distribution $q_N(\Phi_N^c)$, which is derived from the primitive distributions.

**Semantic Prompt Sampling and Mixing.** We treat the learned distributions as priors over the prompt space, and Monte Carlo sampling is used to sample from the enhanced distribution of attributes, objects, and compositions. By integrating all sampled results with original textual prompts, the prompt space is substantially expanded, enabling a more comprehensive characterization of the underlying semantic distribution and enhancing the model's robustness and generalization to unseen compositions. Specifically, the textual prefix representations for each branch are denoted as $[p_{i,1}^a, \ldots, p_{i,m}^a], [p_{j,1}^o, \ldots, p_{j,m}^o]$ and $[p_{k,1}^c, \ldots, p_{k,m}^c]$. We then draw $L$ Monte Carlo sampling from the enhanced distribution $q_N(\Phi_N)$ to obtain $\gamma_l^a, \gamma_l^o, \gamma_l^c, l = 1, 2, \ldots, L$, which represent sampled attribute, object and composition vectors. The mixing process can be shown as:

$$\mathbf{P}_{i,l}^a = [p_{i,1}^a + \gamma_l^a, p_{i,2}^a + \gamma_l^a, \ldots, p_{i,m}^a + \gamma_l^a, \mathbf{v}_i^a], \tag{17}$$

$$\mathbf{P}_{j,l}^o = [p_{j,1}^o + \gamma_l^o, p_{j,2}^o + \gamma_l^o, \ldots, p_{j,m}^o + \gamma_l^o, \mathbf{v}_j^o], \tag{18}$$

$$\mathbf{P}_{k,l}^c = [p_{k,1}^c + \gamma_l^c, p_{k,2}^c + \gamma_l^c, \ldots, p_{k,m}^c + \gamma_l^c, \mathbf{v}_{k,a}^c, \mathbf{v}_{k,o}^c]. \tag{19}$$

Here, $\mathbf{P}_{i,l}^a$, $\mathbf{P}_{j,l}^o$ and $\mathbf{P}_{k,l}^c$ denote the text prompts obtained by mixing the attribute, object and composition prompts with the $l$-th sample. To guarantee correct gradient flow through discrete sampling, the optimization process utilizes the reparameterization trick [25].

**Cross-modal Similarity Score.** The branch-specific textual prompts from Eq. 17–19 are encoded by the text encoder (Eq. 1) to yield textual features $\mathbf{t}_{i,l}^a$, $\mathbf{t}_{j,l}^o$, and $\mathbf{t}_{k,l}^c$. Given an image feature, we compute its similarity to $L$ sampled textual embeddings and take the average across the $L$ samples to obtain the final similarity score for the attribute, object, and composition branches.

Table 1: **Quantitative results** (§4.2) on MIT-States [17], UT-Zappos [65] and C-GQA [42] within *CW* setting.

| *Closed-World* Method | Backbone | MIT-States | | | | UT-Zappos | | | | C-GQA | | | |
|---|---|---|---|---|---|---|---|---|---|---|---|---|---|
| | | Seen↑ | Unseen↑ | HM↑ | AUC↑ | Seen↑ | Unseen↑ | HM↑ | AUC↑ | Seen↑ | Unseen↑ | HM↑ | AUC↑ |
| CLIP [50] ICML2021 | ViT-L | 30.2 | 46.0 | 26.1 | 11.0 | 15.8 | 49.1 | 15.6 | 5.0 | 7.5 | 25.0 | 8.6 | 1.4 |
| CoOp [70] IJCV2022 | ViT-L | 34.4 | 47.6 | 29.8 | 13.5 | 52.1 | 49.3 | 34.6 | 18.8 | 20.5 | 26.8 | 17.1 | 4.4 |
| PCVL [60] Arxiv2022 | ViT-L | 48.5 | 47.2 | 35.3 | 18.3 | 64.4 | 64.0 | 46.1 | 32.2 | - | - | - | - |
| CSP [44] ICLR2023 | ViT-L | 46.6 | 49.9 | 36.3 | 19.4 | 64.2 | 66.2 | 46.6 | 33.0 | 28.8 | 26.8 | 20.5 | 6.2 |
| DFSP(i2t) [36] CVPR2023 | ViT-L | 47.4 | 52.4 | 37.2 | 20.7 | 64.2 | 66.4 | 45.1 | 32.1 | 35.6 | 29.3 | 24.3 | 8.7 |
| DFSP(BiF) [36] CVPR2023 | ViT-L | 47.1 | 52.8 | 37.7 | 20.8 | 63.3 | 69.2 | 47.1 | 33.5 | 36.5 | 32.0 | 26.2 | 9.9 |
| DFSP(t2i) [36] CVPR2023 | ViT-L | 46.9 | 52.0 | 37.3 | 20.6 | 66.7 | 71.7 | 47.2 | 36.0 | 38.2 | 32.0 | 27.1 | 10.5 |
| GIPCOL [61] WACV2024 | ViT-L | 48.5 | 49.6 | 36.6 | 19.9 | 65.0 | 68.5 | 48.8 | 36.2 | 31.9 | 28.4 | 22.5 | 7.1 |
| Troika [15] CVPR2024 | ViT-L | 49.0 | 53.0 | 39.3 | 22.1 | 66.8 | 73.8 | 54.6 | 41.7 | 41.0 | 35.7 | 29.4 | 12.4 |
| PLID [4] ECCV2024 | ViT-L | 49.7 | 52.4 | 39.0 | 22.1 | 67.3 | 68.8 | 52.4 | 38.7 | 38.8 | 33.0 | 27.9 | 11.0 |
| ProLT [18] AAAI2024 | ViT-L | 49.1 | 51.0 | 38.2 | 21.1 | 66.0 | 70.1 | 49.4 | 36.1 | 39.5 | 32.9 | 27.7 | 11.0 |
| CDS-CZSL [34] CVPR2024 | ViT-L | 50.3 | 52.9 | 39.2 | 22.4 | 63.9 | 74.8 | 52.7 | 39.5 | 38.3 | 34.2 | 28.1 | 11.1 |
| **BAYECZSL (Ours)** | ViT-L | $\mathbf{51.7}_{\pm0.5}$ | $\mathbf{51.8}_{\pm0.4}$ | $\mathbf{39.6}_{\pm0.2}$ | $\mathbf{22.5}_{\pm0.2}$ | $\mathbf{67.6}_{\pm1.0}$ | $\mathbf{76.1}_{\pm1.1}$ | $\mathbf{57.6}_{\pm0.7}$ | $\mathbf{45.4}_{\pm0.5}$ | $\mathbf{41.0}_{\pm0.3}$ | $\mathbf{35.5}_{\pm0.2}$ | $\mathbf{30.4}_{\pm0.1}$ | $\mathbf{12.8}_{\pm0.1}$ |

### 3.4 TRAINING AND INFERENCE

**Training.** Based on Eq. 6, we transform the similarity scores into probability distributions and compute the cross-entropy to obtain the final loss function $\mathcal{L}_{\text{aoc}}$. We introduce a Bayesian regularization term $\mathcal{L}_{\text{p}}$ (Eq. 16) to better model the distribution uncertainty. The overall loss function is defined as:

$$\mathcal{L} = \mathcal{L}_{\text{aoc}} + \mathcal{L}_{\text{p}}. \tag{20}$$

**Inference.** During inference, the test image is fed into BAYECZSL to obtain the prediction scores for the attribute $p(a_i \mid x)$, the object $p(o_i \mid x)$, and the composition $p(c_k \mid x)$. The final compositional class is then predicted by integrating the predictions from all three branches:

$$\hat{c} = \underset{c_k \in \mathcal{C}_{\text{test}}}{\arg\max} \ p(c_k \mid x) + p(a_i \mid x) \cdot p(o_j \mid x). \tag{21}$$

## 4 EXPERIMENT

### 4.1 EXPERIMENTAL SETUP

**Datasets.** We conduct experiments on three CZSL benchmarks: MIT-States [17], UT-Zappos [65], and C-GQA [42]. MIT-States consists of 53,753 images representing canonical scenes, encompassing 245 objects and 115 attributes, which together give rise to 1,962 distinct attribute-object compositions. UT-Zappos comprises 29,126 shoe images, categorized into 12 distinct objects and 16 material attributes, resulting in a total of 116 attribute–object compositions. C-GQA contains 39,298 images annotated with 7,732 attribute–object compositions, encompassing 413 distinct attributes and 674 distinct objects. More details are provided in Table 5 (*cf*. §A in Appendix).

**Evaluation Metric.** We follow the evaluation protocol of prior works [42, 48, 34, 15], plotting the unseen-seen accuracy curve with seen accuracy on the X-axis and unseen accuracy on the Y-axis under varying scalars, and computing the Area Under the Curve (AUC). We also report the best Harmonic Mean (HM), best-Seen accuracy (Seen) and best-Unseen accuracy (Unseen). Moreover, AUC is prioritized, as it provides a more comprehensive assessment of the model's performance.

**Implementation Details.** BAYECZSL is based on the pretrained CLIP ViT-L/14 model [50]. For open-world evaluation, we adopt the post-training calibration strategy [44] to filter out infeasible compositions. For fairness, following the existing training setup in prior works [15, 4], our optimization setup uses Adam optimizer with a learning rate of $5 \times 10^{-5}$ for MIT-States with 10 epochs, $1.5 \times 10^{-4}$ for UT-Zappos with 15 epochs, and $1 \times 10^{-5}$ for C-GQA with 15 epochs. For data augmentation, we apply random horizontal flipping and cropping to a resolution of $224 \times 224$. Similar to the CLIP-based prompt learning methods such as Coop [67] and Troika [15], the total number of tokens fed into the CLIP text encoder in our approach is 77. The number of Monte Carlo samples $L$ is set to 12, and the number of reversible mapping layers $N$ for distribution augmentation is set to 15. Further implementation details are provided in §D of Appendix.

**Reproducibility.** BAYECZSL is implemented in PyTorch and trained on one NVIDIA RTX 3090 GPU with a 24GB memory. Training and inference are conducted on the same machine.

Table 2: **Quantitative results** (§4.2) on MIT-States [17], UT-Zappos [65] and C-GQA [42] within *OW* setting.

| Open-World Method | Backbone | MIT-States Seen↑ | Unseen↑ | HM↑ | AUC↑ | UT-Zappos Seen↑ | Unseen↑ | HM↑ | AUC↑ | C-GQA Seen↑ | Unseen↑ | HM↑ | AUC↑ |
|---|---|---|---|---|---|---|---|---|---|---|---|---|---|
| CLIP [50][ICML2021] | ViT-L | 30.1 | 14.3 | 12.8 | 3.0 | 15.7 | 20.6 | 11.2 | 2.2 | 7.5 | 4.6 | 4.0 | 0.3 |
| CoOp [70][IJCV2022] | ViT-L | 34.6 | 9.3 | 12.3 | 2.8 | 52.1 | 31.5 | 28.9 | 13.2 | 21.0 | 4.6 | 5.5 | 0.7 |
| PCVL [60][Arxiv2021] | ViT-L | 48.5 | 16.0 | 17.7 | 6.1 | 64.6 | 44.0 | 37.1 | 21.6 | - | - | - | - |
| CSP [44][ICLR2023] | ViT-L | 46.3 | 15.7 | 17.4 | 5.7 | 64.1 | 44.1 | 38.9 | 22.7 | 28.7 | 5.2 | 6.9 | 1.2 |
| DFSP(i2t) [36][CVPR2023] | ViT-L | 47.2 | 18.2 | 19.1 | 6.7 | 64.3 | 53.8 | 41.2 | 26.4 | 35.6 | 6.5 | 9.0 | 2.0 |
| DFSP(BiF) [36][CVPR2023] | ViT-L | 47.1 | 18.1 | 19.2 | 6.7 | 63.5 | 57.2 | 42.7 | 27.6 | 36.4 | 7.6 | 10.6 | 2.4 |
| DFSP(t2i) [36][CVPR2023] | ViT-L | 47.5 | 18.5 | 19.3 | 6.8 | 66.8 | 60.0 | 44.0 | 30.3 | 38.3 | 7.2 | 10.4 | 2.4 |
| GIPCOL [61][WACV2024] | ViT-L | 48.5 | 16.0 | 17.9 | 6.3 | 65.0 | 45.0 | 40.1 | 23.5 | 31.6 | 5.5 | 7.3 | 1.3 |
| Troika [15][CVPR2024] | ViT-L | 48.8 | 18.7 | 20.1 | 7.2 | 66.4 | 61.2 | 47.8 | 33.0 | 40.8 | 7.9 | 10.9 | 2.7 |
| PLID [4][ECCV2024] | ViT-L | 49.1 | 18.7 | 20.0 | 7.3 | 67.6 | 55.5 | 46.6 | 30.8 | 39.1 | 7.5 | 10.6 | 2.5 |
| CDS-CZSL [34][CVPR2024] | ViT-L | 49.4 | 21.8 | 22.1 | 8.5 | 64.7 | 61.3 | 48.2 | 32.3 | 37.6 | 8.2 | 11.6 | 2.7 |
| **BAYECZSL (Ours)** | ViT-L | **50.2**±0.4 | **18.9**±0.3 | **20.8**±0.2 | **7.6**±0.2 | **69.5**±1.2 | **62.2**±1.0 | **49.7**±0.6 | **35.3**±0.7 | **43.9**±0.3 | **8.4**±0.2 | **11.7**±0.2 | **3.1**±0.1 |

Table 3: **A set of ablation studies** on UT-Zappos [65] and MIT-States [17] within *CW* setting (§4.3).

| Method | UT-Zappos Seen↑ | Unseen↑ | HM↑ | AUC↑ | MIT-States Seen↑ | Unseen↑ | HM↑ | AUC↑ |
|---|---|---|---|---|---|---|---|---|
| BASELINE | 67.2±0.9 | 73.6±0.8 | 55.4±0.6 | 42.6±0.6 | 45.6±0.4 | 52.9±0.5 | 37.3±0.1 | 20.4±0.3 |
| BPD | 67.2±1.2 | 74.7±0.9 | 55.6±0.5 | 43.1±0.7 | 49.2±0.5 | 51.5±0.3 | 38.1±0.1 | 21.3±0.1 |
| BPD + CDS | **67.9**±1.0 | 73.4±0.8 | 57.0±0.4 | 43.7±0.6 | 49.9±0.6 | 51.8±0.2 | 38.6±0.2 | 21.8±0.1 |
| BPD + CDS + TDE | 67.6±1.0 | **76.1**±1.1 | **57.6**±0.7 | **45.4**±0.5 | **51.7**±0.5 | 51.8±0.4 | **39.6**±0.2 | **22.5**±0.2 |

(a) BAYECZSL Components

| Attribute | Object | Composition | UT-Zappos Seen↑ | Unseen↑ | HM↑ | AUC↑ |
|---|---|---|---|---|---|---|
| | | | 67.2 | 73.6 | 55.4 | 42.6 |
| ✓ | | | 65.4 | 74.1 | 56.7 | 42.7 |
| | ✓ | | 65.8 | 73.3 | 56.5 | 42.9 |
| ✓ | ✓ | | 67.2 | **74.7** | 55.7 | 43.1 |
| ✓ | ✓ | ✓ | **68.0** | 73.4 | **57.0** | **43.7** |

(b) Distribution of Three Branches

## 4.2 COMPARISON WITH STATE-OF-THE-ARTS

**Performance on *CW* Setting.** As summarized in Table 1, under *CW* setting, BAYECZSL outperforms recent state-of-the-art (SOTA) CZSL methods across all datasets (*i.e.*, MIT-States [17], UT-Zappos [65], and C-GQA [42]), and sets a new SOTA. In particular, BAYECZSL improves HM by **+0.3**, **+3.0**, and **+1.0** and AUC by **+0.4**, **+3.7**, and **+0.4** on the three datasets. BAYECZSL also achieves the highest accuracy on both seen and unseen accuracies across the UT-Zappos. It can be observed that BAYECZSL significantly improves classification accuracy by exploring a larger prompt space.

**Performance on *OW* Setting.** In *OW* setting illustrated in Table 2, the results show that BAYECZSL consistently delivers strong performance across all three datasets [17, 65, 42]. Especially on UT-Zappos, BAYECZSL achieves the best performance of **49.7** (**+4.0%**) HM and **35.3** (**+7.0%**) AUC. The learned prompt distributions effectively explore intra-primitive diversity, covering a more diverse range of prompts and enabling BAYECZSL to excel in the expansive search space of the open-world scenario. For complex datasets with large-scale and highly diverse attributes, such as C-GQA, our model achieves the best performance of **11.7** (**+7.3%**) HM and **3.1** (**+14.8%**) AUC.

## 4.3 ABLATION STUDY

**Key Component Analysis.** We first study the efficacy of our core idea and model designs, which is summarized in Table 3a. We conduct ablation studies based on the BAYECZSL baseline, where three key components are incrementally incorporated. Specifically, BPD denotes *Bayesian-induced Primitive Distribution Learning* component, CDS represents *Compositional Distribution Synthesis* component, and TDE refers to *Three-path Distribution Enhancement* component. In the second row, only the primitive prompt branch is modeled as a probability distribution on top of the baseline. This core component yields clear gains in both HM and AUC. In the third row, we further incorporate CDS, which builds an auxiliary compositional distribution upon primitive distributions, resulting in additional improvements, *e.g.*, **+1.4** HM on UT-Zappos [65] and **+0.5** HM on MIT-States [17]. Finally, enhancing textual prompt distributions of three branches via TDE leads to a substantial improvement in performance, *e.g.*, **+1.7** AUC on UT-Zappos and **+0.7** AUC on MIT-States.

**Primitive Distribution Learning.** We next evaluate the effectiveness of learning image-specific probabilistic distributions in primitive (*i.e.*, attribute and object) and compositional branches in Table 3b. It is worth emphasizing that the results reported in Table 3b are obtained without applying

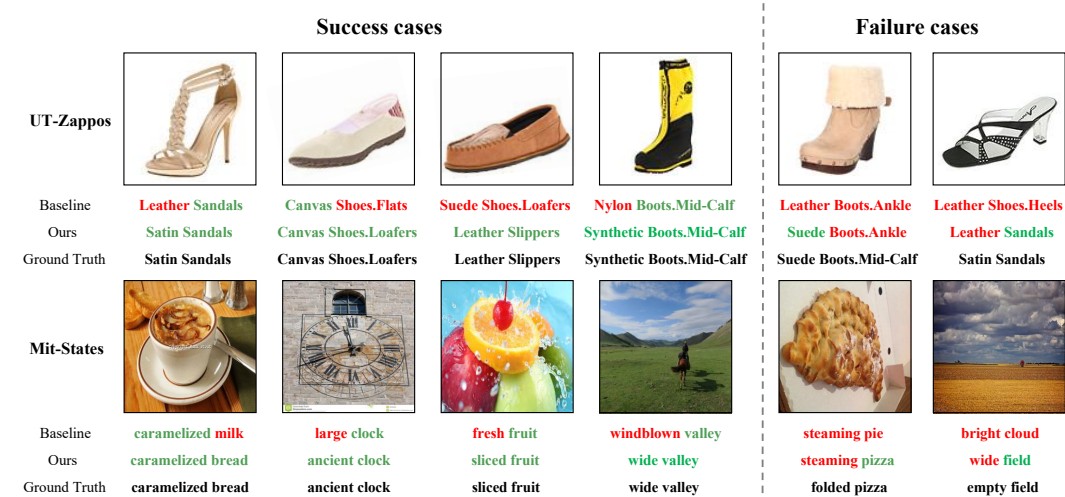

Figure 3: We show top-1 predictions of BAYECZSL in comparison with baseline. Correct predictions are highlighted in green, and incorrect predictions in red.

Table 4: **Ablation of the Hyperparameters** on UT-Zappos [65] within *CW* setting (§4.3).

| $N = 5$ | UT-Zappos | | | | $N = 10$ | UT-Zappos | | | |
|---|---|---|---|---|---|---|---|---|---|
| | Seen↑ | Unseen↑ | HM↑ | AUC↑ | | Seen↑ | Unseen↑ | HM↑ | AUC↑ |
| $L = 3$ | 63.6 | 72.5 | 51.2 | 38.0 | $L = 3$ | 69.3 | 75.9 | **56.4** | **44.3** |
| $L = 6$ | 69.4 | 73.6 | 52.5 | 40.5 | $L = 6$ | 63.4 | 76.1 | 53.4 | 39.8 |
| $L = 9$ | 65.3 | 71.2 | **55.3** | **41.1** | $L = 9$ | 66.0 | 70.4 | 51.2 | 38.2 |
| $L = 12$ | 67.4 | 73.7 | 53.4 | 40.9 | $L = 12$ | 66.6 | 72.8 | 55.8 | 42.1 |
| $L = 15$ | 65.8 | 73.7 | 53.0 | 39.6 | $L = 15$ | 67.5 | 74.3 | 54.5 | 42.4 |
| (a) Ablation with $N = 5$ | | | | | (b) Ablation with $N = 10$ | | | | |

| $N = 15$ | UT-Zappos | | | | $N = 20$ | UT-Zappos | | | |
|---|---|---|---|---|---|---|---|---|---|
| | Seen↑ | Unseen↑ | HM↑ | AUC↑ | | Seen↑ | Unseen↑ | HM↑ | AUC↑ |
| $L = 3$ | 64.0 | 72.5 | 52.2 | 39.1 | $L = 3$ | 65.5 | 75.8 | **54.4** | **41.2** |
| $L = 6$ | 64.0 | 75.4 | 52.3 | 39.5 | $L = 6$ | 61.6 | 74.6 | 50.9 | 36.5 |
| $L = 9$ | 68.0 | 75.4 | 56.7 | 44.0 | $L = 9$ | 67.8 | 73.7 | 52.3 | 40.2 |
| $L = 12$ | 67.6 | 76.1 | **57.6** | **45.4** | $L = 12$ | 61.8 | 74.3 | 49.9 | 35.2 |
| $L = 15$ | 67.8 | 75.7 | 56.4 | 44.0 | $L = 15$ | 63.7 | 73.7 | 53.8 | 39.8 |
| (c) Ablation with $N = 15$ | | | | | (d) Ablation with $N = 20$ | | | | |

distribution enhancement, and thus solely reflect the effect of introducing probabilistic modeling into different branches. For one of the primitive branches, we remove the distribution construction and instead use the original soft prompt. The results in Row 2 and 3 show that applying probabilistic modeling to either the attribute or the object branch individually already yields noticeable performance gains. Row 4 demonstrates that modeling both primitive branches simultaneously enables BAYECZSL to capture better intra-primitive diversity, thereby improving classification accuracy. Moreover, modeling probability distributions across all three branches leads to further gains in both HM and AUC. This indicates that jointly applying probabilistic modeling to attribute, object, and compositional levels allows the model to achieve more comprehensive coverage of the prompt space.

**Sensitivity Analysis on Monte Carlo Sampling Number $L$ and Reversible Mapping Layer Number $N$.** Table 4 presents comprehensive ablations on the sampling number $L$ and the number of reversible mapping layers $N$ on the UT-Zappos dataset. Overall, both hyperparameters consistently improve the model's performance when set within a reasonable range, but choosing values that are either excessively small or overly large results in significantly degraded performance, exhibiting a typical "sweet-spot" behavior commonly observed in hyperparameter tuning.

When fixing $N$, different values of $L$ significantly influence the quality of distribution characterization. A small $L$ results in insufficient sampling, preventing the model from adequately exploring the latent space. Conversely, an excessively large $L$ introduces unnecessary noise, which harms performance. Across configurations with $N = 5, 10, 15, 20$, we consistently observe optimal HM and AUC around $L = 12$. When fixing $L$, the number of reversible mapping layers $N$ also impacts the expressiveness of the model. A shallow mapping limits the capacity to model complex distributions, whereas an

overly deep mapping introduces redundant parameters and training instability, ultimately reducing performance. Results across all tested $L$ values show that performance peaks around $N = 15$.

These findings suggest that a moderate sampling scale (e.g., $L = 12$) and a reasonable mapping depth (e.g., $N = 15$) provide the most balanced and stable performance, while deviations from this region lead to noticeable degradation. More detailed analyses can be found in Appendix §D.

### 4.4 QUALITATIVE RESULTS

In Fig. 3, we visualize both the successful and failed cases of our BAYECZSL, as well as examples from the baseline model without the Bayesian-induced learning. For instance, BAYECZSL is able to correctly adjust the material of "leather" to "satin" in the UT-Zappos dataset, and "fresh" to "sliced" in the MIT-States dataset. This demonstrates that BAYECZSL can capture intra-primitive diversity and more comprehensive relationships between primitives and their compositions. The last two columns in the figure show failure cases, where the images contain ambiguous primitive cues, causing BAYECZSL to make incorrect predictions at the composition level. Nevertheless, the predicted compositions still provide a reasonable interpretation of the image content. More success and failure cases are provided in §F of Appendix.

## 5 CONCLUSION

In this paper, we propose BAYECZSL, a novel Bayesian-induced framework for Compositional Zero-Shot Learning that learns probability distributions over primitive prompts to capture intra-primitive diversity and semantic uncertainty. Then, by aggregating the probabilistic distributions of the attribute and object branches into a unified compositional prompt space, BAYECZSL captures the rich semantic relationships between primitive concepts and their compositions. Moreover, Three-path Distribution Enhancement module is introduced to transform initial distributions into expressive ones via invertible mappings, facilitating diverse prompt sampling from these complex distributions. Experiments on three datasets confirm the superiority of our Bayesian-induced framework.

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

## A    DETAILED STATISTICS ON DATA SPLITTING

We experiment with three real-world CZSL benchmarks: MIT-States [17], UT-Zappos [65], and C-GQA [42]. The MIT-States dataset is constructed by collecting images from diverse real-world scenes and manually annotating them with the corresponding attributes and objects. The MIT-States dataset consists of 53,753 images depicting canonical scenes, encompassing 245 objects and 115 attributes, which together give rise to 1,962 attribute–object compositions. Following the standard split, these compositions are divided into 1,262 seen compositions for training, and 300 seen as well as 400 unseen compositions for validation and testing. The UT-Zappos dataset consists of 29,126 shoe images, categorized into 12 distinct object classes and annotated with 16 material attributes, resulting in a total of 116 attribute–object compositions. The dataset is divided into 83 seen compositions for train, 15 seen and 15 unseen compositions for validation, and 18 seen and 18 unseen compositions for test. Unlike datasets that involve relatively simple attribute–object compositions, the UT-Zappos dataset primarily focuses on subtle variations in shoe materials, which poses significant challenges for compositional models. The C-GQA dataset encompasses common attribute concepts together with object concepts encountered in everyday contexts, making it the most comprehensive benchmark for CZSL. It consists of 39,298 images annotated with 413 distinct attributes and 674 distinct objects, forming over 9,500 attribute–object compositions. Following the standard split, the dataset includes 5,592 seen compositions for training, 1,252 seen and 1,040 unseen compositions for validation, and 888 seen and 923 unseen compositions for testing. The detailed data split statistics is provided in Table 5.

Table 5: Summary of data splits (§5) of MIT-States [17], UT-Zappos [65], and C-GQA [42].

| Dataset | Composition | | | Train | | Validation | | Test | |
|---|---|---|---|---|---|---|---|---|---|
| | $|\mathcal{A}|$ | $|\mathcal{O}|$ | $|\mathcal{A}|\times|\mathcal{O}|$ | $|\mathcal{C}_s|$ | $|\mathcal{X}|$ | $|\mathcal{C}_s|/|\mathcal{C}_u|$ | $|\mathcal{X}|$ | $|\mathcal{C}_s|/|\mathcal{C}_u|$ | $|\mathcal{X}|$ |
| MIT-States [17] | 115 | 245 | 28,175 | 1,262 | 30,338 | 300 / 300 | 10,420 | 400 / 400 | 12,995 |
| UT-Zappos [65] | 16 | 12 | 192 | 83 | 22,998 | 15 / 15 | 3,214 | 18 / 18 | 2,914 |
| C-GQA [42] | 413 | 674 | 278,362 | 5,592 | 26,920 | 1,252 / 1,040 | 7,280 | 888 / 923 | 5,098 |

## B    LLM USAGE STATEMENT

We utilized a large language model (*e.g.*, ChatGPT) to assist in refining the wording of specific sentences and paragraphs within this paper. The sole purpose of this tool is to enhance the clarity and readability of the text. Importantly, LLM is not employed in any core research processes, including method design, the generation of experimental results, or the formulation of research conclusions.

## C BASELINE MODEL DETAILS

**Visual Representation Learning.** Following [34, 15], for a given image $x \in \mathbb{R}^{H \times W \times 3}$, we employ the CLIP image encoder $E_{\text{img}}$ to divide it into $N_p = HW/P^2$ patches, where (P, P) is the resolution of each patch. We extend the PETL technique to the visual domain, where it is instantiated in the form of an adapter [12]. The patches are transformed into a sequence of patch tokens, augmented with a pre-trained [CLS] token, while pre-trained positional embeddings are incorporated to preserve spatial information. We employ the [CLS] token as the image representation $\mathbf{x}^c$, and subsequently adopt an attribute adapter $D_a$ and an object adapter $D_o$, both implemented as MLPs, to disentangle $\mathbf{x}^c$ into the attribute feature $\mathbf{x}^a$ and the object feature $\mathbf{x}^o$.

**Prompt Representation Learning.** Following [15], we adopt a three-path paradigm to construct the prompts. For each attribute-object composition $c_{i,j} = \langle a_i, o_j \rangle$, we construct attribute prompt $\mathbf{P}_i^a = [p_{i,1}^a, \ldots, p_{i,m}^a, \mathbf{v}_i^a]$, object prompt $\mathbf{P}_j^o = [p_{j,1}^o, \ldots, p_{j,m}^o, \mathbf{v}_j^o]$ and composition prompt $\mathbf{P}_k^c = [p_{k,1}^c, \ldots, p_{k,m}^c, \mathbf{v}_{k,a}^c, \mathbf{v}_{k,o}^c]$. All prompts are learnable vectors, and the prompt prefixes $p_{i,1:m}^a$, $p_{j,1:m}^o$, and $p_{k,1:m}^c$ are initialized with *"a photo of"*. Subsequently, these prompts are fed into the frozen text encoder of CLIP [50] to obtain the corresponding prompt features.

**Feasibility Calibration for Open-World Setting.** Following [39, 44], post-training feasibility calibration is employed to eliminate infeasible compositions that may occur during open-world evaluation. This procedure operates under the assumption that semantically similar objects are more likely to share compatible attributes, whereas dissimilar objects are unlikely to exhibit such commonality. Therefore, given a candidate pair $c = \langle a, o \rangle$, similarities between the objects can be computed as:

$$\rho_o(a, o) = \max_{\hat{o} \in \mathcal{O}^{ae}} \frac{\phi(o) \cdot \phi(\hat{o})}{\|\phi(o)\| \|\phi(\hat{o})\|}. \tag{22}$$

Here, $\mathcal{O}^{ae}$ denotes the set of objects that co-occur with attribute $a$ in the seen compositions. The function $\phi(\cdot)$ represents the embedding mapping that projects each primitive into a pre-trained embedding space, instantiated with GloVe embeddings [47].Analogously, attribute similarities $\rho_a(a, o)$ are computed following the same procedure. Finally, the feasibility score for a composition $(a, o)$ is obtained by aggregating the two similarity measures using a mean pooling function $\mu$:

$$\rho(a, o) = \mu\big(\rho_o(a, o), \rho_a(a, o)\big). \tag{23}$$

Finally, infeasible compositions are pruned by retaining only those whose feasibility score satisfies $\rho(a, o) > T$ on the validation set. The final prediction is then obtained as

$$\hat{c} = \underset{c_{i,j} \in \mathcal{C}^{tgt}, \, \rho(a_i, o_j) > T}{\operatorname{argmax}} p(c_k \mid x) + p(a_i \mid x) \cdot p(o_j \mid x). \tag{24}$$

## D MORE QUANTITATIVE RESULTS

**Sample Number and Mapping Layers.** We conducted experiments on UT-Zappos, evaluating different combinations of sampling numbers and mapping layers (Fig. 4).

For $N = 5$, performance initially increases with the number of samples, peaking at $L = 9$, and then slightly decreases. This suggests that a moderate number of samples helps, but excessive sampling introduces redundancy. Overall, performance remains limited, indicating that $N = 5$ layers constrain model expressiveness, suitable mainly for simpler tasks.

For $N = 10$, performance decreases as sample size increases, with the best results at $L = 3$. This indicates that deeper layers without sufficient capacity may not benefit from larger sample sizes, and excessive samples can lead to overfitting and computational redundancy.

For $N = 15$, both HM and AUC improve with more samples, reaching the optimum at $L = 12$. The deeper mapping layers provide richer feature representations, allowing the model to effectively utilize more samples to enhance feature learning, stabilize training, and improve generalization.

For $N = 20$, performance fluctuates, with the best results at $L = 3$. Excessively deep layers increase model capacity, which can cause instability and overfitting, especially with limited data. Smaller sample sizes balance learning capacity and training data diversity, leading to optimal performance.

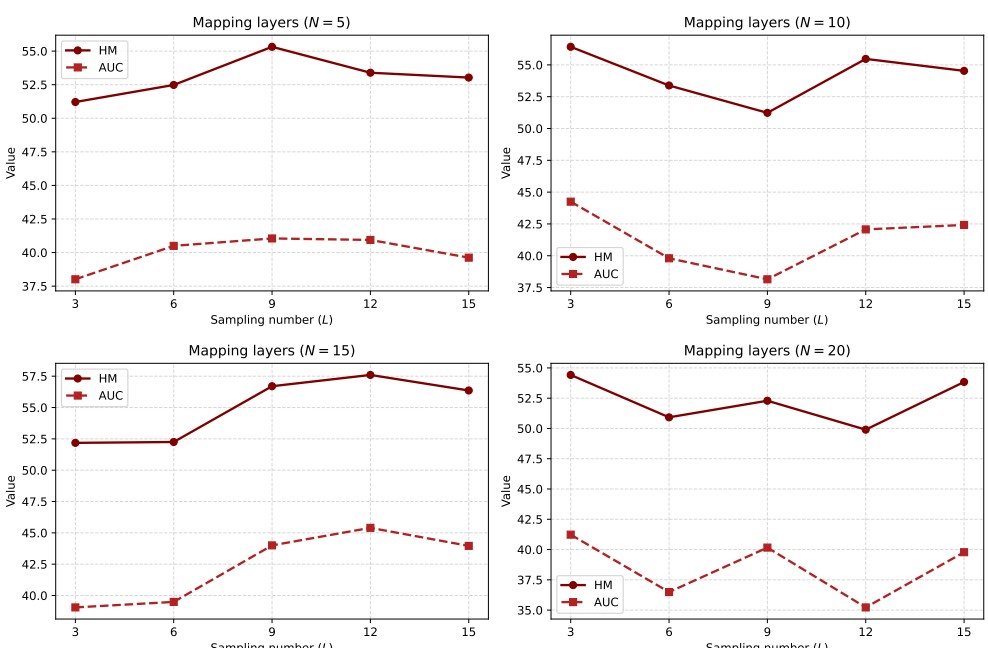

Figure 4: AUC and HM under Different Mapping Layers and Sampling Numbers.

**Efficiency and Performance Analysis of BAYECZSL.** We conducted a systematic efficiency analysis of the proposed BAYECZSL method and compared it with the state-of-the-art Troika [15] and our baseline model, with results summarized in Table 6. BAYECZSL has 14.9M trainable parameters. This lightweight advantage primarily stems from the compact design of our probabilistic prompt distribution modeling, which maintains a high degree of parameter sharing and avoids the introduction of additional large networks. Although BAYECZSL exhibits slightly higher memory usage (11.7G) and training time (19.6 min) compared to the baseline and Troika, its inference latency is only 25.1 ms, remaining well within acceptable limits for practical deployment without significant impact on efficiency.

In terms of performance, BAYECZSL achieves an AUC of 45.4 on the UT-Zappos dataset, representing an improvement of +2.8 over the baseline and +3.5 over Troika. This substantial performance gain is highly acceptable given the modest additional computational overhead, demonstrating that our probabilistic distribution enhancement framework effectively improves compositional generalization while maintaining strong efficiency. These results indicate that BAYECZSL achieves a superior balance between performance and computational cost, confirming its practical utility in real-world scenarios.

Table 6: Efficiency comparison on UT-Zappos [65]. Here, we report trainable parameters, training time per epoch, and inference speed for each model. See in §D for more details.

| Method | Params↑ | Memory↑ | Training time↑ | Inference Speed↑ | AUC↑ |
|--------|---------|---------|----------------|------------------|------|
| Troika [15] | 21.7M | 9.0G | 15.1min | 22.0ms | 41.9 |
| Baseline | 7.6M | 8.6G | 11.8min | 14.2ms | 42.6 |
| **BAYECZSL (ours)** | 14.9M | 11.7G | 19.6min | 25.1ms | 45.4 |

**More Comparison Results with Existing CZSL Methods.** Apart from CLIP-based approaches, we further compare our proposed BAYECZSL with existing CZSL methods, all of which adopt ResNet18 as the backbone. Evaluations are conducted on three datasets, with the results reported in Table 7 under the closed-world setting and in Table 8 under the open-world setting. It can be observed that, by transferring pre-trained knowledge, CLIP-based methods consistently outperform other CZSL approaches in both settings. Notably, our proposed BAYECZSL achieves state-of-the-art performance across all cases.

Table 7: **More comparison results**(§D) on MIT-States [17], UT-Zappos [65] and C-GQA [42] within *closed world* setting.

| Closed-World | MIT-States | | | | UT-Zappos | | | | C-GQA | | | |
|---|---|---|---|---|---|---|---|---|---|---|---|---|
| Method | Seen↑ | Unseen↓ | HM↑ | AUC↑ | Seen↑ | Unseen↑ | HM↑ | AUC↑ | Seen↑ | Unseen↓ | HM↑ | AUC↑ |
| *Traditional vision-based methods* | | | | | | | | | | | | |
| AoP [43] | 14.3 | 17.4 | 9.9 | 1.6 | 59.8 | 54.2 | 40.8 | 25.9 | 17.0 | 5.6 | 5.9 | 0.7 |
| LE+ [41] | 15.0 | 20.1 | 10.7 | 2.0 | 53.0 | 61.9 | 41.0 | 25.7 | 18.1 | 5.6 | 6.1 | 0.8 |
| TMN [48] | 20.2 | 20.1 | 13.0 | 2.9 | 58.7 | 60.0 | 45.0 | 29.3 | 23.1 | 6.5 | 7.5 | 1.1 |
| SymNet [33] | 24.2 | 25.2 | 16.1 | 3.0 | 49.8 | 57.4 | 40.4 | 23.4 | 26.8 | 10.3 | 11.0 | 2.1 |
| CompCos [38] | 25.3 | 24.6 | 16.4 | 4.5 | 59.8 | 62.5 | 43.1 | 28.1 | 28.1 | 11.2 | 12.4 | 2.6 |
| CGE [42] | 28.7 | 25.3 | 17.2 | 5.1 | 56.8 | 63.6 | 41.2 | 26.4 | 28.1 | 10.1 | 11.4 | 2.3 |
| Co-CGE [39] | 27.8 | 25.2 | 17.5 | 5.1 | 58.2 | 63.3 | 44.1 | 29.1 | 29.3 | 11.9 | 12.7 | 2.8 |
| SCEN [31] | 29.9 | 25.2 | 18.4 | 5.3 | 63.5 | 63.1 | 47.8 | 32.0 | 28.9 | 12.1 | 12.4 | 2.9 |
| CVGAE [1] | 28.5 | 25.5 | 18.2 | 5.3 | 65.0 | 62.4 | 49.8 | 34.6 | 28.2 | 11.9 | 13.9 | 2.8 |
| CANet [58] | 29.0 | 26.2 | 17.9 | 5.4 | 61.0 | 66.3 | 47.3 | 33.1 | 30.0 | 13.2 | 14.5 | 3.3 |
| CAPE [23] | 30.5 | 26.2 | 19.1 | 5.8 | 60.4 | 67.4 | 45.5 | 31.3 | 32.9 | 15.6 | 16.3 | 4.2 |
| *CLIP-based methods* | | | | | | | | | | | | |
| CLIP [50] | 30.2 | 46.0 | 26.1 | 11.0 | 15.8 | 49.1 | 15.6 | 5.0 | 7.5 | 25.0 | 8.6 | 1.4 |
| CoOp [70] | 34.4 | 47.6 | 29.8 | 13.5 | 52.1 | 49.3 | 34.6 | 18.8 | 20.5 | 26.8 | 17.1 | 4.4 |
| PCVL [60] | 48.5 | 47.2 | 35.3 | 18.3 | 64.4 | 64.0 | 46.1 | 32.2 | - | - | - | - |
| CSP [44] | 46.6 | 49.9 | 36.3 | 19.4 | 64.2 | 66.2 | 46.6 | 33.0 | 28.8 | 26.8 | 20.5 | 6.2 |
| DFSP(i2t) [36] | 47.4 | 52.4 | 37.2 | 20.7 | 64.2 | 66.4 | 45.1 | 32.1 | 35.6 | 29.3 | 24.3 | 8.7 |
| DFSP(BiF) [36] | 47.1 | 52.8 | 37.7 | 20.8 | 63.3 | 69.2 | 47.1 | 33.5 | 36.5 | 32.0 | 26.2 | 9.9 |
| DFSP(t2i) [36] | 46.9 | 52.0 | 37.3 | 20.6 | 66.7 | 71.7 | 47.2 | 36.0 | 38.2 | 32.0 | 27.1 | 10.5 |
| GIPCOL [61] | 48.5 | 49.6 | 36.6 | 19.9 | 65.0 | 68.5 | 48.8 | 36.2 | 31.9 | 28.4 | 22.5 | 7.1 |
| Troika [15] | 49.0 | **53.0** | 39.3 | 22.1 | 66.8 | 73.8 | 54.6 | 41.7 | 41.0 | **35.7** | 29.4 | 12.4 |
| PLID [4] | 49.7 | 52.4 | 39.0 | 22.1 | 67.3 | 68.8 | 52.4 | 38.7 | 38.8 | 33.0 | 27.9 | 11.0 |
| **BAYECZSL (Ours)** | **51.7** | 51.8 | **39.6** | **22.5** | **67.6** | **76.1** | **57.6** | **45.4** | **41.0** | 35.5 | **30.4** | **12.8** |

Table 8: **More comparison results**(§D) on MIT-States [17], UT-Zappos [65] and C-GQA [42] within *open world* setting.

| Open-World | MIT-States | | | | UT-Zappos | | | | C-GQA | | | |
|---|---|---|---|---|---|---|---|---|---|---|---|---|
| Method | Seen↑ | Unseen↑ | HM↑ | AUC↑ | Seen↑ | Unseen↑ | HM↑ | AUC↑ | Seen↑ | Unseen↓ | HM↑ | AUC↑ |
| *Traditional vision-based methods* | | | | | | | | | | | | |
| AoP [43] | 16.6 | 5.7 | 4.7 | 0.7 | 50.9 | 34.2 | 29.4 | 13.7 | - | - | - | - |
| LE+ [41] | 14.2 | 2.5 | 2.7 | 0.3 | 60.4 | 36.5 | 30.5 | 16.3 | 19.2 | 0.7 | 1.0 | 0.1 |
| TMN [48] | 12.6 | 0.9 | 1.2 | 0.1 | 55.9 | 18.1 | 21.7 | 8.4 | - | - | - | - |
| SymNet [33] | 21.4 | 7.0 | 5.8 | 0.8 | 53.3 | 44.6 | 34.5 | 18.5 | 26.7 | 2.2 | 3.3 | 0.4 |
| CompCos [38] | 25.4 | 10.0 | 8.9 | 1.6 | 59.3 | 46.8 | 36.9 | 21.3 | 28.4 | 1.8 | 2.8 | 0.4 |
| CGE [42] | 29.6 | 4.0 | 4.9 | 0.7 | 58.8 | 46.5 | 38.0 | 21.5 | 28.3 | 1.3 | 2.2 | 0.3 |
| Co-CGE [39] | 26.4 | 10.4 | 10.1 | 2.0 | 60.1 | 44.3 | 38.1 | 21.3 | 28.7 | 1.6 | 2.6 | 0.4 |
| KG-SP [22] | 28.4 | 7.5 | 7.4 | 1.3 | 61.8 | 52.1 | 42.3 | 26.5 | 31.5 | 2.9 | 4.7 | 0.8 |
| CVGAE [1] | 27.3 | 9.9 | 10.0 | 1.8 | 58.6 | 48.4 | 41.7 | 22.2 | 26.6 | 2.9 | 6.4 | 0.7 |
| *CLIP-based methods* | | | | | | | | | | | | |
| CLIP [50] | 30.1 | 14.3 | 12.8 | 3.0 | 15.7 | 20.6 | 11.2 | 2.2 | 7.5 | 4.6 | 4.0 | 0.3 |
| CoOp [70] | 34.6 | 9.3 | 12.3 | 2.8 | 52.1 | 31.5 | 28.9 | 13.2 | 21.0 | 4.6 | 5.5 | 0.7 |
| PCVL [60] | 48.5 | 16.0 | 17.7 | 6.1 | 64.6 | 44.0 | 37.1 | 21.6 | - | - | - | - |
| CSP [44] | 46.3 | 15.7 | 17.4 | 5.7 | 64.1 | 44.1 | 38.9 | 22.7 | 28.7 | 5.2 | 6.9 | 1.2 |
| DFSP(i2t) [36] | 47.2 | 18.2 | 19.1 | 6.7 | 64.3 | 53.8 | 41.2 | 26.4 | 35.6 | 6.5 | 9.0 | 2.0 |
| DFSP(BiF) [36] | 47.1 | 18.1 | 19.2 | 6.7 | 63.5 | 57.2 | 42.7 | 27.6 | 36.4 | 7.6 | 10.6 | 2.4 |
| DFSP(t2i) [36] | 47.5 | 18.5 | 19.3 | 6.8 | 66.8 | 60.0 | 44.0 | 30.3 | 38.3 | 7.2 | 10.4 | 2.4 |
| GIPCOL [61] | 48.5 | 16.0 | 17.9 | 6.3 | 65.0 | 45.0 | 40.1 | 23.5 | 31.6 | 5.5 | 7.3 | 1.3 |
| Troika [15] | 48.8 | 18.7 | 20.1 | 7.2 | 66.4 | 61.2 | 47.8 | 33.0 | 40.8 | 7.9 | 10.9 | 2.7 |
| PLID [4] | 49.1 | 18.7 | 20.0 | 7.3 | 67.6 | 55.5 | 46.6 | 30.8 | 39.1 | 7.5 | 10.6 | 2.5 |
| **BAYECZSL (Ours)** | **50.2** | **18.9** | **20.8** | **7.6** | **69.5** | **62.2** | **49.7** | **35.3** | **43.9** | **8.4** | **11.7** | **3.1** |

## E    SEMANTIC PROMPT SAMPLING AND MIXING

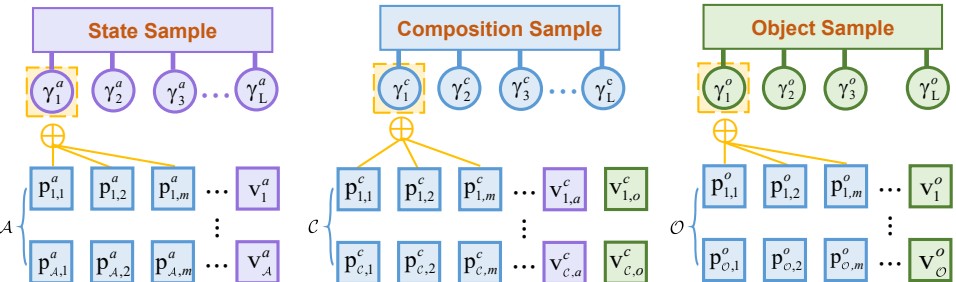

Figure 5: Illustration of Monte Carlo sampling and prompt mixing for expanding the prompt space.

**Process of Semantic Prompt Sampling and Mixing.** As shown in Fig. 5, we perform Monte Carlo sampling for each of the three branches and then fuse each sample with its corresponding prompt prefix.

## F    MORE QUALITATIVE VISUALIZATION

**More Case Study.** We provide additional success and failure cases of our method BAYECZSL across three CZSL benchmarks, *i.e.*, MIT-States [17] in Fig. 6, UT-Zappos [65] in Fig. 7 and C-GQA [42] in Fig. 8. We also compare our approach BAYECZSL with baseline without Bayesian-induced framework. As shown in the figure, by modeling the intra-primitive variance through Bayesian learning, the model can achieve more comprehensive coverage of the prompt space and generalize to unseen compositions. For more fine-grained primitives, such as rich colors, textures, and appearances, the model can make accurate predictions. Even when the same attribute appears with different visual expressions across various compositions, the model can still clearly distinguish the differences.

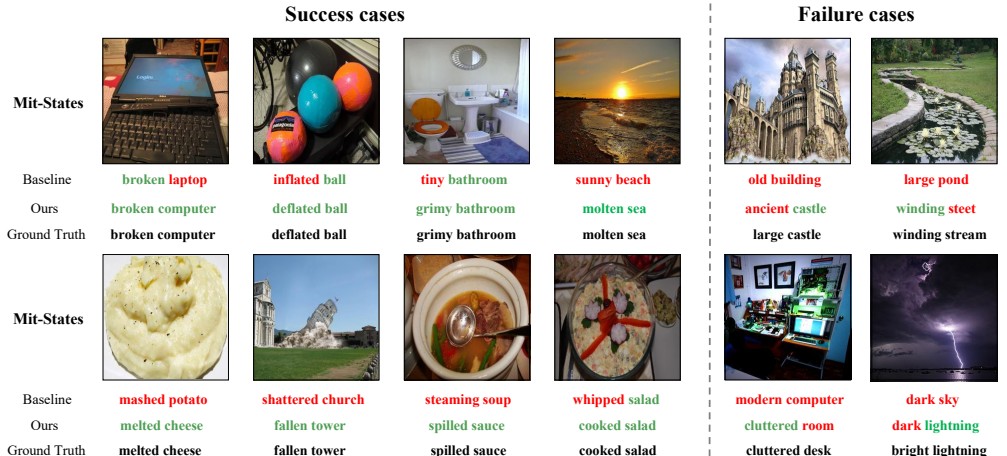

Figure 6: Additional case studies on Mit-States [17] are presented, where BAYECZSL is compared with the baseline that does not include Bayesian-induced framework. Correct predictions are highlighted in green, and incorrect ones in red.

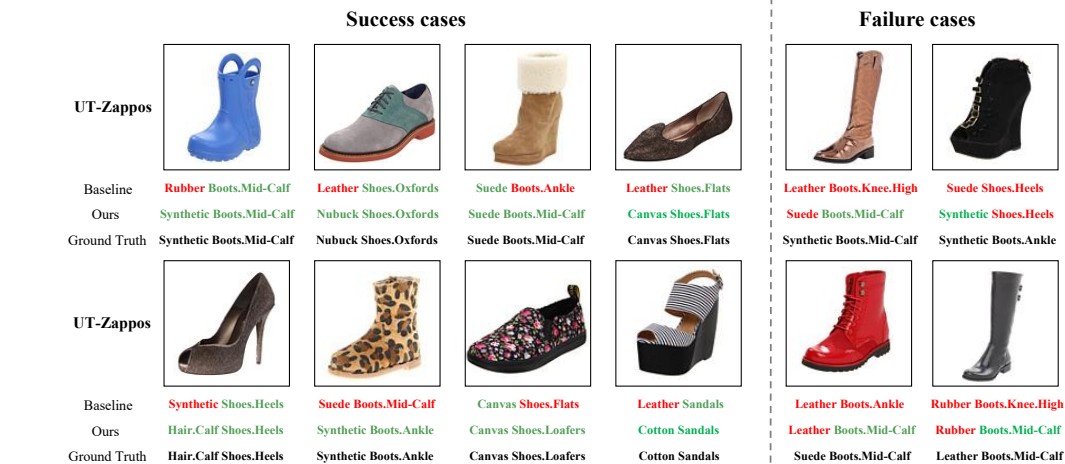

Figure 7: Additional case studies on UT-Zappos [65] are presented, where BAYECZSL is compared with the baseline that does not include Bayesian-induced framework. Correct predictions are highlighted in green, and incorrect ones in red.

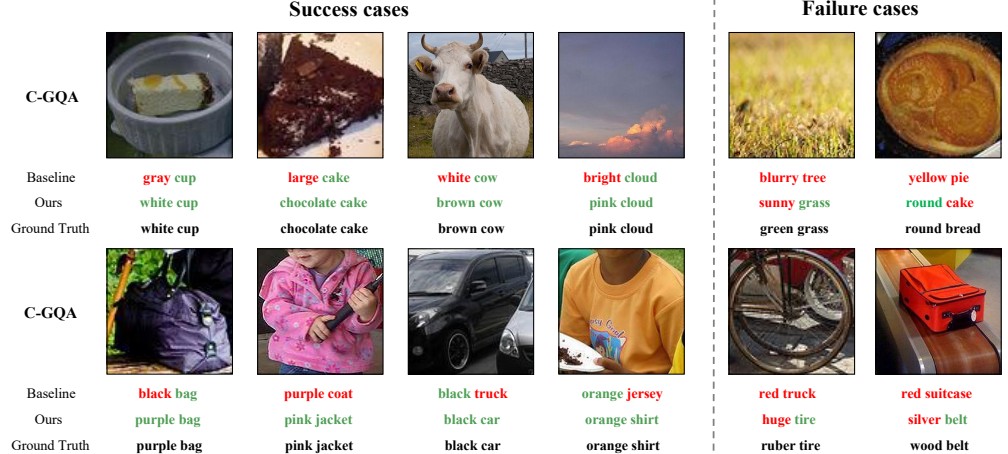

Figure 8: Additional case studies on C-GQA [42] are presented, where BAYECZSL is compared with the baseline that does not include Bayesian-induced framework. Correct predictions are highlighted in green, and incorrect ones in red.

## G IMPACT OF THE HYPERPARAMETER $\beta$

The ablation study on UT-Zappos [65] regarding the influence of the hyperparameter $\beta$ is presented in Table 9. We conducted a sensitivity analysis on the loss weights for the attribute, object, and compositional branches, $(\beta_a, \beta_o, \beta_c)$ within the range [0.5, 2]. The results indicate that increasing any branch weight to 2 or decreasing it to 0.5 significantly reduces both HM and AUC, demonstrating that the model performance is relatively sensitive to the loss weights of the three branches.

Table 9: Effect of $(\beta_a, \beta_o, \beta_c)$ on UT-Zappos [65].

| $(\beta_a, \beta_o, \beta_c)$ | Seen↑ | Unseen↑ | HM↑ | AUC↑ |
|---|---|---|---|---|
| (1, 1, 0.5) | 66.2 | 74.4 | 55.6 | 42.9 |
| (1, 1, 2) | 66.5 | 74.5 | 55.7 | 43.0 |
| (1, 0.5, 1) | 67.2 | 73.6 | 55.0 | 42.0 |
| (1, 2, 1) | 65.2 | 73.2 | 54.0 | 40.0 |
| (0.5, 1, 1) | 66.2 | 73.6 | 54.4 | 41.3 |
| (2, 1, 1) | **70.0** | 75.3 | 57.6 | 44.5 |
| (1, 1, 1) | 67.6 | **76.1** | **57.6** | **45.4** |

# H  PERFORMANCE COMPARISON OF COCOOP AND BAYECZSL

As shown in Table 10, we report the average AUC and standard error for CoCoOp [69] and BAYECZSL over 5 random seeds on UT-Zappos [65]. CoCoOp is a conditional variant of CoOp that generates image-specific bias vectors using visual information and adds them to the prompt vocabulary, thereby improving few-shot object classification performance. In this study, we examine whether such image-conditioned prompts can also enhance performance in compositional zero-shot learning tasks. In contrast, BAYECZSL introduces modules such as BPD, CDS, and TDE to more effectively model compositional distributions. Experimental results show that BAYECZSL outperforms CoCoOp across all three datasets. These findings indicate that, although incorporating image-conditioned prompts can provide some performance improvement, the compositional modeling design of BAYECZSL can significantly boost AUC, validating the effectiveness of our method in compositional zero-shot learning tasks.

Table 10: Performance comparison with CoCoOp [69] on UT-Zappos [65].

| Method | MIT-States | UT-Zappos | C-GQA |
|---|---|---|---|
| CoCoOp [69] | $11.3_{\pm0.6}$ | $18.8_{\pm1.1}$ | $4.2_{\pm0.1}$ |
| Ours | $22.5_{\pm0.2}$ | $45.4_{\pm0.5}$ | $12.8_{\pm0.1}$ |

# I  COMPOSITIONAL DISTRIBUTION FUSION STRATEGY

**Analysis of Fusion Strategies for Compositional Representations.** The results of different fusion strategies are shown in table 11. If a simple weighted geometric mean were used, the attribute and object branches would be assigned the same confidence. However, in most compositions, the contributions of attributes and objects are generally different. The inverse-variance weighted Gaussian fusion assigns larger weights to branches with lower uncertainty, meaning that more reliable information (smaller variance) contributes more. This fusion strategy naturally reflects the uncertainty of semantic components while suppressing bias introduced by noisy signals, resulting in more stable and generalizable compositional representations.

Table 11: Ablation study about fusion strategies on UT-Zappos [65].

| Method | Seen | Unseen | HM | AUC |
|---|---|---|---|---|
| Weighted Geometric Mean | 65.6 | 74.1 | 55.1 | 41.8 |
| Ours | **67.6** | **76.1** | **57.6** | **45.4** |

**Impact of Fusion on Compositional Prompt Expressiveness.** If the distribution is extracted directly from the compositional branch without any fusion, the model cannot fully leverage the complementary information from the attribute and object branches. This leads to less expressive compositional prompt distributions, insufficient coverage of the diversity of primitive concepts, and limited generalization to unseen attribute–object compositions. As shown in the table 12, the fusion strategy improves performance.

Table 12: Ablation study about without fusion on UT-Zappos [65].

| Method | Seen | Unseen | HM | AUC |
|---|---|---|---|---|
| Direct Compositional Branch | 69.7 | 74.9 | 57.0 | 44.5 |
| Ours | **67.6** | **76.1** | **57.6** | **45.4** |

As shown in Table 13, we conducted an experiment by removing the disentangling MLP and the CDS module in our method, and directly predicting a compositional posterior distribution from the combined representation $x^c$ obtained via Eq. 2. This simplified structure no longer models the individual attribute/object posteriors, nor performs precision-weighted fusion. We carried out this ablation study on the UT-Zappos dataset for comparison.

Table 13: Ablation study about without MLP and CDS on UT-Zappos [65].

| Method | Seen ↑ | Unseen ↑ | HM ↑ | AUC ↑ |
|---|---|---|---|---|
| Without MLP and CDS | 66.4 | 73.7 | 54.7 | 41.4 |
| Ours | **67.6** | **76.1** | **57.6** | **45.4** |

## J  DISTRIBUTION ENHANCEMENT STRATEGIES

To further enhance the model's expressive power, we introduce the TDE module on top of BPD. By stacking invertible mappings (similar to normalizing flows), TDE transforms the simple diagonal Gaussian into a more complex distribution. This design allows the model to maintain the simplicity of the initial assumption while modeling nonlinear dependencies and complex input-conditioned structures, effectively "shifting complexity upstream." As shown in Table 14, we conducted experiments on UT-Zappos using three alternative posterior modeling strategies: full-covariance, mixture posteriors, and standard normalizing flows. Although we carefully tuned these methods to achieve reasonable performance, they still underperform TDE. This indicates that the diagonal-Gaussian assumption in BPD has limited impact on the results, and the invertible mappings in TDE are sufficient to enhance posterior expressiveness.

Table 14: Ablation study about different distribution enhancement strategies on UT-Zappos [65].

| Method | Seen ↑ | Unseen ↑ | HM ↑ | AUC ↑ |
|---|---|---|---|---|
| Full-Covariance | **67.9** | 72.9 | 56.0 | 42.2 |
| Mixture Posteriors | 66.6 | 70.0 | 54.3 | 40.5 |
| Normalizing Flows | 65.6 | 72.8 | 54.8 | 41.5 |
| TDE | 67.6 | **76.1** | **57.6** | **45.4** |

## K  EVALUATION RESULTS ACROSS DIFFERENT PRE-TRAINED MODELS

As shown in Table 15, we conducted experiments using the ViT-B backbone, which demonstrate that our method exhibits strong robustness and superiority.

Table 15: Performance comparison on CLIP-ViT-B backbone on UT-Zappos [65].

| Method | Backbone | Seen ↑ | Unseen ↑ | HM ↑ | AUC ↑ |
|---|---|---|---|---|---|
| ViT-B | Baseline | 61.0 | 62.9 | 45.1 | 31.9 |
| ViT-B | Ours | **65.7** | **67.5** | **51.3** | **37.3** |

## L  GENERALIZATION TO HIGHER-ORDER COMPOSITIONS

Previous work (CSP) [44] introduced another challenging dataset: AAO-MIT-States, a subset derived from MITStates to evaluate the higher-order compositional learning ability in the form of attribute-attribute-object (AAO) compositions. This approach allows us to accommodate multi-attribute objects without altering the original disentanglement framework, while preserving the Bayesian modeling of uncertainty. As shown in Table 16, our framework achieves strong performance on this dataset.

Table 16: Quantitative comprasion results on AAO-MIT-States.

| Model | Accuracy |
|---|---|
| CLIP | 62.7 |
| CSP | $72.6_{\pm 0.4}$ |
| Ours | $74.9_{\pm 0.7}$ |

