# OpenReview forum: "Bayesian Primitive Distributing for Compositional Zero-shot Learning"
_ICLR.cc/2026/Conference — Submitted to ICLR 2026_

### Official Review · Reviewer_Hjnp · 2025-11-01

**Soundness:** 2
**Presentation:** 2
**Contribution:** 2
**Rating:** 4
**Confidence:** 4

**Summary:**

The paper introduces BAYECZSL, a novel Bayesian-induced framework that learns distributions over primitive textual prompts. The authors observe that the existing CZSL works use a single deterministic textual prompt for each primitive concept and its composition, which is insufficient to capture variations within the compositions; for example, old in old dog is different from old town. Furthermore, they notice that prior work ignores the rich relational structure between the primitives and compositions. BAYECZSL addresses these limitations through the following steps. First, it learns a probability distribution over primitive concepts to better represent intra-primitive diversity and reduce overfitting. Then, it uses compositional distributional synthesis to aggregate the learned probability distributions into a unified compositional prompt space. Then, to model more complex distributions, it uses a three-path distribution enhancement module to transform the initial prompt and composition distributions into flexible distributions using a sequence of invertible mappings. Finally, it draws multiple Monte Carlo samples from the distributions and mixes them with the original prompt representations to improve generalization. The results on the CZSL benchmarks show that BAYECZSL improves performance over the state-of-the-art methods.

**Strengths:**

The paper is easy to follow and well-written.

The proposed method is well-motivated. The method outperforms prior work on compositional zero-shot learning datasets.

The ablations in Section 4.3 are quite helpful in understanding the contributions of BAYECZSL.

**Weaknesses:**

**Method**

- The core components (variational posteriors, Gaussian fusion, normalizing-flow enhancement, and Monte Carlo sampling) are established Bayesian/VI tools. Novelty is rather modest since it mainly adapts these techniques to CZSL prompt distributions.

- The approach uses MLP-based disentanglers to obtain attribute and object features and to parameterize their base posteriors. The compositional posterior is obtained via inverse-variance weighted fusion rather than being learned directly. A simpler alternative is to get a composition posterior from $x^{c}$ (Eq. 2) without CDS and additional disentanglers; an ablation here would be helpful.

- The method is close to CoCoOp [a]. CoCoOp adds additional information about the image, in the form of a meta token, to the text prompts, thereby improving performance. Although this paper is included in the related work, it is not compared in the results section. Including them as baselines or explaining why they are not directly comparable would make the evaluation fairer.

- The framework assumes single attribute-object compositions and does not evaluate multi-attribute or multi-object cases. Suppose there is an object with multiple attributes at test time (e.g., small white cat), the framework could potentially sample vectors from the same enhanced distributions for multiple attributes rather than treating them as separate attributes. This could lead to a performance drop when integrated into the prompt vectors. Including experiments on attribute-attribute-object settings [b] would strengthen the paper.

**Architecture.**

All the experiments are limited to the CLIP ViT-L/14 model. It would be great if the authors could include experiments with additional CLIP models and other models such as BLIP, etc. It is also unclear from the paper if their method will even work with vision-language models that use a decoder instead of the bi-encoder architecture seen in CLIP (Figure 2).

**Impact**

While the paper shows positive, albeit small, improvements over prior methods on the compositional zero-shot learning datasets, my concern is that the paper is too task-specific.
The paper makes strong assumptions about the types of compositions it can handle, i.e., the method can only handle attribute-object compositions. This severely limits the impact of the paper.

**Minor suggestions**

Lines 97-99: It would be great if the authors could explicitly say that they are reporting relative improvement in performance. At first glance, it appeared to be an absolute improvement.

**References**

[a] Conditional Prompt Learning for Vision-Language Models, CVPR 22.

[b] Learning to Compose Soft Prompts for Compositional Zero-Shot Learning, ICLR 23.

**Questions:**

In addition to the questions listed in the weaknesses section, here are a few more questions.

**Clarification for Misc. claim**

- Lines 61-63: What does “cross-branch synergies” mean? Could you explain that in simpler words?

- In lines 60-61, the authors say that prior work ignores the rich relational structure between the primitives and their compositions. Could you clarify what this sentence means and how BAYECZSL understands the relational structure of the concepts? In addition, could you also discuss Appendix D in more detail in the main paper? The plots suggest the model's performance can drop below the best numbers reported in Table 1. I’d like to understand the trade-off between the number of Monte Carlo samples and performance.

**Error bars**

Since you are averaging over $L$ prompts, could you also include the error bars for the method in the results section?

---

> ### Author Response · Authors · 2025-11-23
> **Point-to-Point Response to Reviewer Hjnp (1/3)**
>
> Thank you so much for the valuable time and constructive feedback for us to improve the paper. We provide point-to-point response below.
>
> ---
>
> **Q1: Novelty of our BAYECZSL.**
>
> **A1:** We respectfully disagree. We argue that our work represents a novel architectural paradigm in Compositional Zero-Shot Learning (CZSL) filed far beyond a reuse of prior techniques such as variational posteriors and Gaussian fusion. Our proposed framework is well-motivated by the application domain (*i.e.*, CZSL), and the motivation and reasoning behind it are clearly articulated. In addition, our method also introduces concrete technical innovations. Below, we clarify the key distinctions between our work and existing approaches in terms of the motivation and technical design.
>
> **1. Motivation: Paradigm Shift from Deterministic to Probabilistic Primitives**
>
> Existing CLIP-based CZSL methods typically learn one single deterministic textual prompt for each primitive or compositional label. This approach is oversimplified and ignores the inherent "**intra-primitive diversity**" (*e.g.*, the semantic meaning of "old" varies significantly between "old dog" and "old town"). In contrast, we are the first to model image-specific primitive prompts as learnable probability distributions rather than static vectors. By applying Bayesian inference specifically to the primitive level, we capture the semantic uncertainty and natural variability of attributes and objects.  This is a domain-specific architectural choice designed to solve the problem of context-dependency in CZSL, rather than a generic application of VI.
>
> **2. Technical Design**
>
> - **Compositional distribution modeling strategy**
>
>   Existing CLIP-based CZSL methods treat attribute, object, and composition prompts independently or fuse them using deterministic operations (*e.g.*, simple concatenation or averaging). These methods fail to model how the uncertainty of one primitive affects the final composition. In contrast, we introduce the Compositional Distribution Synthesis (CDS) module, which aggregates the learned probability distribution of both attribute and object branches to form the compositional prompt space.  This module allows us to explicitly capture the semantic relationships and the propagation of uncertainty between the primitives.
>
> - **Standard diagonal assumptions vs. expressive flow-based Posteriors**
>
>     Standard Bayesian Approaches rely on diagonal Gaussian assumptions for the posterior due to computational convenience. However, this "vanilla" application limits the model's ability to capture the complex semantic correlations inherent in high-dimensional prompt spaces. In contrast, we go beyond the "modest adaptation" of standard tools by integrating the Three-path Distribution Enhancement (TDE) module utilizing invertible mappings. Unlike standard VI, our framework applies a sequence of invertible mappings to transform simple initial distributions into highly expressive, non-Gaussian posteriors. This represents a technical deepening of the Bayesian framework specifically tailored to ensure the semantic fidelity of primitives.
>
>
> **Q2: Ablation study regarding composition posterior.**
>
> **A2:** Thanks for your suggestion. The “simpler alternative” mentioned refers to removing the disentangling MLP and the CDS module in our method, and directly predicting a compositional posterior distribution from the combined representation $x^c$ obtained via Eq. (2).  We conducted this ablation on the UT-Zappos dataset for comparison. The experiment is added in Appendix Sec.I and Table 13.
>
> | Method| Seen ↑ | Unseen ↑ | HM ↑ | AUC ↑ |
> |:-|:------:|:-:|:-:|:-:|
> | Without MLP and CDS| 66.4 | 73.7| 54.7 | 41.4  |
> | Ours| **67.6**| **76.1**|**57.6**|**45.4**|
>
> The experimental results show a significant performance drop, indicating that directly learning the compositional posterior from $x^c$ fails to capture the semantic shifts of attributes and objects in different compositions.
>
> We further performed an ablation where the disentangling MLP is retained but CDS is removed, the distribution is directly extracted from the compositional branch. The experiment is added in Appendix Sec.I and Table 12.
>
> | Method| Seen ↑ | Unseen ↑ | HM ↑ | AUC ↑ |
> |:-|:-:|:-:|:-:|:-:|
> | Direct Compositional Branch| 69.7| 74.9|57.0|44.5|
> | Ours|**67.6**|**76.1**| **57.6** | **45.4**  |
>
> The results indicate that the model cannot leverage the complementary information between the attribute and object branches, which limits the expressiveness of the compositional prompt distribution, thereby restricting generalization to unseen attribute–object combinations.
>
> In contrast, our method explicitly obtains the semantic distributions of attributes and objects via the disentangler and constructs the compositional posterior with the CDS. This enables the model to capture the semantic relationships, which cannot be replaced by the simplified alternatives.

---

> > ### Author Response · Authors · 2025-11-23
> > **Point-to-Point Response to Reviewer Hjnp (2/3)**
> >
> > **Q3: Comparison with CoCoOp.**
> >
> > **A3:** Good suggestion! We add a direct comparison with CoCoOp as suggested. As shown in the table below, our method consistently outperforms CoCoOp on MIT-States, UT-Zappos, and C-GQA in terms of AUC, with the results reported as the average over 5 random seeds along with the standard errors. This inclusion ensures a fairer evaluation by directly comparing with a closely related baseline that incorporates image-conditioned prompts. The experiment is added in Appendix Sec.H and Table 10.
> >
> > | Method   | MIT-States          | UT-Zappos          | C-GQA             |
> > |----------|-------------------|------------------|-----------------|
> > | CoCoOp   | 11.3±0.6| 18.8±1.1 | 4.2±0.1 |
> > | Ours     | **22.5**±0.2 | **45.4**±0.5 | **12.8**±0.1 |
> >
> > Although incorporating image-conditioned prompts can provide some performance improvement, the compositional modeling design of BAYECZSL can significantly boost AUC, validating the effectiveness of our method in CZSL.
> >
> > **Q4: More evaluation on AAO-MIT-States.**
> >
> > **A4:** Thank you for the insightful comment. Our current method disentangles the original image features into a single attribute feature and an object feature, which is suitable for single-attribute objects. In the case of multi-attribute objects, multiple attributes must be sampled from the same attribute feature, which may introduce dependencies among attributes and potentially degrade the quality of the compositional representation.
> >
> > To address this, we propose re-encoding or re-fusing the disentangled attribute features:First, leveraging independent subspaces for attributes, the single attribute feature can be expanded into multiple attribute distributions via linear transformations. Next, for each target attribute, independent vectors are sampled from these expanded distributions to ensure greater independence among attributes in the compositional representation. Finally, these attribute vectors are combined with the object feature to form the final compositional representation for prompt generation or downstream tasks.
> >
> > This approach allows us to accommodate multi-attribute objects without altering the original disentanglement framework, while preserving the Bayesian modeling of uncertainty. The experiment on AAO-MIT-States, a subset derived from MIT-States in the form of attribute-attribute-object (AAO) compositions, is added in Appendix Sec.L and Table 16.
> >
> > | Model          | Accuracy |
> > |:-------------:|:--------:|
> > | CLIP           |  62.7    |
> > | CSP            |  72.6±0.4    |
> > | Ours |  **74.9**±0.7   |
> >
> > As seen, the results verifies the effectiveness of our method under the multi-attribute setting.
> >
> >
> > **Q5: Evaluation results across different pre-trained models.**
> >
> > **A5:** Thank you for your suggestion. To further demonstrate the robustness of our method, we conduct additional experiments using the ViT-B backbone. The experiment is added in Appendix Sec.K and Table 15. The results validate the effectiveness of our approach, showing consistent performance gains across various CLIP model scales. Although our current experiments are primarily based on the CLIP-style bi-encoder architecture, the Bayesian modeling of compositional distributions is model-agnostic and can, in principle, be adapted to vision-language models that use a decoder. Adapting to such models may require modifications in how the prompt vectors are integrated. We plan to conduct a comprehensive evaluation on decoder-based architectures such as BLIP in future work.
> >
> > | Method   | Backbone |Seen ↑ | Unseen ↑ | HM ↑ | AUC ↑ |
> > |:--------:|:--------|:------:|:--------:|:----:|:-----:|
> > | Baseline| ViT-B |  61.0 | 62.9   | 45.1 | 31.9  |
> > | Ours|ViT-B    |  **65.7**   | **67.5**     | **51.3** | **37.3**  |
> >
> > **Q6: Impact.**
> >
> > **A6:** We fully understand your concerns regarding the task-specific nature of our method. Currently, our approach primarily assumes that each composition consists of a single attribute and a single object. However, our experiments demonstrate that the method consistently improves performance across all settings and datasets we evaluated. Furthermore, we also observe effectiveness on multi-attribute objects, indicating that our framework can handle more complex attribute combinations to a certain extent.
> >
> > Although the current version does not yet cover more complex compositions, we plan to extend our experiments in future work, including evaluations on compositional zero-shot video datasets and more intricate multi-attribute and multi-object scenarios. This will allow us to test the model’s performance in more complex compositional and temporal settings.

---

> > > ### Author Response · Authors · 2025-11-23
> > > **Point-to-Point Response to Reviewer Hjnp (3/3)**
> > >
> > > **Q7: Updated performance evaluation.**
> > >
> > > **A7:** Thank you for your suggestion. The reported performance in the paper refers to relative improvements. We make the following modifications in the revision:
> > >
> > >     Lines 97-99:  Concretely, on the CW setting, BAYECZSL exceeds the current state-of-the-art methods by up to +8.9% and +3.2% relative AUC improvement on UT-Zappos and C-GQA. Under the more challenging OW setting, BAYECZSL still urpasses the best CLIP-based method by up to +7.0% and +14.8% relative AUC improvement on UT-Zappos and C-GQA.
> > >
> > > **Q8: Clarify the mean of cross-branch synergies.**
> > >
> > > **A8:** Sorry for this confusion. We use 'cross-branch synergy' to describe the process where the attribute and object branches mutually support each other—leveraging the learned distributions from both to produce a compositional distribution that is more robust than either branch alone. We have clarified this definition in the revision.
> > >
> > > **Q9: Clarification.**
> > >
> > > **A9:** Regarding the “rich relational structure between primitives and their compositions” mentioned in lines 60–61, we refer to the inherent semantic dependencies between attributes and objects within a compositional context. In BAYECZSL, we capture this structure by extracting image-conditioned attribute and object posteriors via disentanglers. Crucially, these posteriors are integrated using inverse-variance weighted fusion to form the final compositional prompt distribution. This mechanism explicitly preserves the relational structure by allowing the composition to be dynamically determined by the confidence of each primitive branch.
> > >
> > > Additionally, we discuss Appendix Sec.D in more detail in the main text. Specifically, Appendix Sec.D presents the effects of the number of samples (N) and the number of mapping layers (L) on model performance in the UT-Zappos dataset. The results indicate that:
> > >
> > > - An appropriate combination of N and L can improve performance. For example, N = 15 and L = 12 achieve optimal results.
> > > - Too few or too many samples, or mapping layers that are too shallow or too deep, can lead to performance degradation or instability, mainly due to insufficient capacity, redundancy, or overfitting.
> > >
> > > These analyses help explain the results in Table 1 and illustrate the trade-offs between sampling quantity and model depth.  We incorporate the results and discussion into Table 4 and Sec. 4.3.
> > >
> > >
> > > **Q10: Error bars.**
> > >
> > > **A10:** Thank you for your suggestion. To further enhance reproducibility, we conduct additional experiments with 5 or more random seeds and report both the mean and standard deviation/confidence intervals, providing more robust quantitative support and facilitating faithful reproduction by other researchers. We updated Tables 1, 2 and 3 regarding main comparisons and ablations to report the mean and standard deviation.
> > >
> > > ---
> > >
> > > We appreciate again your thoughtful review and we hope we addressed your concerns. Please let us know if you'd like any further information.

---

> > > > ### Comment · Reviewer_Hjnp · 2025-11-26
> > > > **Response to the reviewers**
> > > >
> > > > Thank you for addressing most of my questions and concerns, and also including additional results in the appendix. I will update my score to 6.
> > > >
> > > > I believe the paper is quite thorough and shows strong results on the benchmarks. However, I am less convinced about the long-term impact this work could have. Nevertheless, the BAYECZSL improves on prior work in compositional zero-shot learning and would be of interest to a focused audience.

---

> > > > > ### Author Response · Authors · 2025-11-27
> > > > > **Response to Reviewer Hjnp**
> > > > >
> > > > > Dear Reviewer Hjnp,
> > > > >
> > > > > Thank you again for your positive feedback. We are glad that our responses and the revised paper addressed your concerns, and we sincerely appreciate your updated score. Regarding long-term impact, we plan to extend our BAYECZSL to more scenarios and tasks in the future. If you have any further questions or suggestions, please feel free to share them.
> > > > >
> > > > > Sincerely yours,
> > > > >
> > > > > Authors

---

### Official Review · Reviewer_QwoX · 2025-11-01

**Soundness:** 3
**Presentation:** 3
**Contribution:** 3
**Rating:** 6
**Confidence:** 4

**Summary:**

The paper proposes BAYECZSL, a Bayesian-induced framework for compositional zero-shot learning (CZSL) that represents each primitive textual prompt (attribute/object) as a probability distribution, rather than a single deterministic vector like most prior work in recent years. WIthin the proposed approach, these distributions (of attributes and objects) are image-conditioned via variational inference, then synthesized into a compositional prompt distribution using variance-inverse Gaussian fusion; distributions are further made expressive by a three-path distribution enhancement module based on invertible mappings (normalizing-flow–style). Sampling from these distributions yields diverse prompts that are mixed with the soft prompts in a three-path CLIP-based architecture (attribute/object/composition). The loss combines branch cross-entropies with a Bayesian regularizer, and inference fuses composition scores with the product of primitive scores. The authors conduct experiments on standard CZSL benchmarks MIT-States, UT-Zappos, and C-GQA show state-of-the-art results in closed-world and open-world settings. The authors also conduct ablations on the three modules and sampling sensitivity.

**Strengths:**

To the best of my knowledge, the paper’s core idea—modeling attribute/object primitive prompts as image-conditioned distributions, composing them into a compositional distribution via variance-inverse Gaussian fusion is novel for CZSL.

I think this probabilistic framing is clear and technically sound: the objectives are explicit, the inference story is coherent, and the components (BPD, CDS, TDE) are well-motivated. I think the evaluation setup in this paper is correct and consistent with the CZSL literature (closed-world and open-world settings on MIT-States, UT-Zappos, and C-GQA), and the approach shows strong performance overall, with especially robust gains on UT-Zappos. I like that the ablations are clean and isolate each module’s contribution, and the sensitivity analyses over the sampling count L and the mapping depth N are informative rather than cosmetic.

Overall, I think this paper is a good work on compositionality in CZSL with a principled probabilistic formulation and solid empirical support.

**Weaknesses:**

While I have no concerns about the novelty or the methodological soundness of this work, my primary concern is reproducibility. The reported numbers appear to be single-run point estimates without variability, and I do not see evidence of repeated runs or variance statistics. Including results over multiple random seeds (idealy over 5) with mean +/- confidence intervals would materially strengthen the quantitative claims, especially for the main comparisons and ablations. Clearly specifying the random seed policy, sources of stochasticity (e.g., initialization of the flow/TDE, sampling count L, data shuffling), and any early-stopping criteria would also help others reproduce the results faithfully.

**Questions:**

I am wondering if the authors have thought about how sensitive are results to the diagonal-Gaussian residual assumption in BPD? Have the authors tried things like full-covariance, mixture posteriors, or normalizing flows in place of TDE to shift complexity upstream?

---

> ### Author Response · Authors · 2025-11-23
> **Point-to-Point Response to Reviewer QwoX**
>
> Thank you so much for the valuable time and constructive feedback for us to improve the paper. We provide point-to-point response below.
>
> ---
>
> **Q1: Experimental reproducibility.**
>
> **A1:** Following your suggestion, we updated Tables 1, 2 and 3 regarding main comparisons and ablations to report results averaged over 5 random seeds, including both mean and standard deviations to demonstrate stability. To ensure reproducibility, we standardize the training protocol: all models are trained for a fixed 15 epochs without early stopping, and the validation set is strictly reserved for hyperparameter tuning. We clarify that the primary sources of randomness include parameter initialization (*e.g.*, TDE module), Monte-Carlo sampling (size $L$), and data shuffling. The updated results confirm the robustness of our method.
>
>
>
> **Q2.1: Diagonal-Gaussian Residual Assumption in the BPD Module.**
>
> **A2.1:** In the attribute and object branches, we map each primitive’s visual feature to a diagonal Gaussian distribution as the initial assumption for residual modeling. This assumption ensures computational efficiency and training stability, while variational inference (ELBO) is used to learn the primitive distributions, capturing the semantic uncertainty of each primitive. We acknowledge that a simple diagonal Gaussian might be insufficiently expressive. To address this, we integrated the Three-path Distribution Enhancement (TDE) module to transform the initial simple diagonal Gaussian into a highly expressive, non-Gaussian posterior. Our ablation study in Table 3a validates this design.
>
> **Q2.2: Distribution Enhancement via the TDE Module.**
>
> **A2.2:** To further enhance the model’s expressive power, we introduce the TDE module on top of BPD. By stacking invertible mappings (similar to normalizing flows), TDE transforms the simple diagonal Gaussian into a more complex distribution. This design allows the model to maintain the simplicity of the initial assumption while modeling nonlinear dependencies and complex input-conditioned structures, effectively “shifting complexity upstream.”
>
> Additionally, we implemented three alternative posterior modeling strategies, full-covariance, mixture posteriors, and standard normalizing flows, and compared their performance on the UT-Zappos dataset. Related results will be added in Appendix Sec. J and Table 14.
>
> | Method              | Seen ↑ | Unseen ↑ | HM ↑ | AUC ↑ |
> |:-------------------|:------:|:--------:|:----:|:-----:|
> | Full-Covariance     | **67.9**   | 72.9     | 56.0 | 42.2  |
> | Mixture Posteriors  | 66.6   | 70.0     | 54.3 | 40.5  |
> | Normalizing Flows   | 65.6   | 72.8     | 54.8 | 41.5  |
> | TDE             | 67.6   | **76.1**     | **57.6** | **45.4**  |
>
>
> Although we carefully tuned these methods to achieve reasonable performance, they still underperform TDE. This indicates that the diagonal-Gaussian assumption in BPD has limited impact on the results, and the invertible mappings in TDE are sufficient to enhance posterior expressiveness.
>
> In summary, we conclude that the BPD module is not sensitive to the diagonal-Gaussian assumption, and the TDE module effectively increases posterior complexity and expressiveness while maintaining computational efficiency and stability.
>
> ---
>
> We appreciate again your thoughtful review and we hope we addressed your concerns. Please let us know if you'd like any further information.

---

> ### Author Response · Authors · 2025-11-27
> **Looking forward to the discussion**
>
> Dear Reviewer,
>
> We sincerely appreciate the time and effort you have dedicated to reviewing our submission. We have submitted our rebuttal and would like to follow up to inquire whether our responses have sufficiently addressed your concerns.
>
> Please let us know if you have any remaining questions or require additional clarification. We value your feedback and are eager to ensure our work meets the highest standards.
>
> Authors

---

### Official Review · Reviewer_th9S · 2025-11-01

**Soundness:** 4
**Presentation:** 4
**Contribution:** 4
**Rating:** 10
**Confidence:** 5

**Summary:**

This paper proposes BAYECZSL, a Bayesian-induced CZSL framework that learns probability distributions over each primitive textual prompt from a Bayesian perspective. The method explicitly models prompt uncertainty for attributes and objects, then synthesizes compositional distributions through a principled fusion mechanism, and enhances those distributions with invertible mappings. Experiments on three major CZSL benchmarks (MIT-States, UT-Zappos, C-GQA) demonstrate BAYECZSL outperforms existing CZSL methods in both Closed-World and Open-World settings.

**Strengths:**

1) This paper proposes very novel ideas to more effectively tackle the core challenge of intra-primitive semantic diversity in compositional zero-shot learning via Bayesian distribution modeling.
2) The key idea of learning probability distributions over each primitive textual prompt, rather than learning a single deterministic prompt as in prior work, is both theoretically grounded and interesting. In addition, this idea is well-aligned with the core challenge.
3) I am also generally impressed  by the the novel use of primitive distributions to construct a compositional prompt space, and the practical use of distribution enhancement strategy to facilitate diverse prompt sampling.
4) The experimental results are convincing, naturally leading the reader to concur with the authors’ perspective.
5) The paper is exceptionally well-written and a true pleasure to read.

**Weaknesses:**

1) Its better to explain why the variance-inverse weight Gaussian fusion strategy is used in the Compositional Distribution Synthesis module.
2) More extensive ablation experiments on more datasets, such as MIT-States or C-GQA, would improve the experiment part.
3) Its better to analyze the impact of the hyper-parameters $\beta_a,\beta_o,\beta_c$.
4) The model introduces computational overhead compared to single-prompt and simple soft-prompt baselines, given multiple sampling, flows, and synthesizing steps. Its better to analyze training/inference cost, memory consumption, or tradeoffs between performance and complexity.

**Questions:**

How is numerical stability maintained during covariance inversion in the compositional distribution synthesis step? Is regularization ever needed, and does this affect fusion quality?

---

> ### Author Response · Authors · 2025-11-23
> **Point-to-Point Response to Reviewer th9S (1/2)**
>
> Thank you so much for the valuable time and constructive feedback for us to improve the paper. We provide point-to-point response below.
>
> ---
>
> **Q1: The choice of the inverse-variance weighted Gaussian fusion strategy.**
>
> **A1:** Thanks for your comment. We explain our choice of this strategy from the following two aspects:
>
> - **Weighted geometric average alternative:** If a simple weighted geometric mean were used, the attribute and object branches would be assigned the same confidence. However, in most compositions, the contributions of attributes and objects are generally different. The inverse-variance weighted Gaussian fusion assigns larger weights to branches with lower uncertainty, meaning that more reliable information (smaller variance) contributes more. This fusion strategy naturally reflects the uncertainty of semantic components while suppressing bias introduced by noisy signals, resulting in more stable and generalizable compositional representations. We have supplemented our experiments using the weighted geometric average fusion strategy on UT-Zappos, and the results are as follows:
>     | Method                  | Seen | Unseen | HM   | AUC  |
>     |-------------------------|:----:|:------:|:----:|:----:|
>     | Weighted Geometric Mean | 65.6 | 74.1   | 55.1 | 41.8 |
>     | Ours                    | **67.6** | **76.1**  | **57.6** | **45.4** |
>
> - **Without fusion strategy:** If the distribution is extracted directly from the compositional branch without any fusion, the model cannot fully leverage the complementary information from the attribute and object branches. This leads to less expressive compositional prompt distributions, insufficient coverage of the diversity of primitive concepts, and limited generalization to unseen attribute–object compositions. We have additionally included experiments on UT-Zappos for this direct compositional branch extraction, and the results are shown below:
>     | Method                       | Seen | Unseen | HM   | AUC  |
>     |------------------------------|:----:|:------:|:----:|:----:|
>     | Direct Compositional Branch  | 69.7 | 74.9   | 57.0 | 44.5 |
>     | Ours                         | **67.6** | **76.1**  | **57.6** | **45.4** |
>
> The experiment is added in Appendix Sec.I and Table 11, 12. Thanks.
>
> **Q2: Additional Ablation Experiments.**
>
> **A2:** Thank you for your valuable suggestion. In the revised manuscript, we add the module-wise ablation results on the MIT-States dataset into Table 3(a). The results are as follows:
>
> | Method                     | Seen ↑ | Unseen ↑ | HM ↑ | AUC ↑ |
> |---------------------------|:------:|:--------:|:----:|:-----:|
> | Baseline                  | 45.6±0.4   | 52.9±0.5    | 37.3±0.1 | 20.4±0.3  |
> | BPD                       | 49.2±0.5  | 51.5±0.3    | 38.1±0.2 | 21.3±0.1 |
> | BPD + CDS                 | 49.9±0.6| 51.8±0.2  | 38.6±0.2| 21.8±0.1  |
> | BPD + CDS + TDE           | **51.7**±0.5   | **51.8**±0.4 | **39.6**±0.2| **22.5**±0.2|
>
> These results clearly demonstrate the effectiveness of our proposed design. Each component in our framework contributes meaningful improvements, and their combination leads to consistent gains across the major evaluation metrics.
>
> **Q3: Impact of the hyperparameter β.**
>
> **A3:** Regarding the influence of the hyperparameter β, we conducted a sensitivity analysis on the loss weights for the attribute, object, and compositional branches, (βa, βo, βc) within the range [0.5, 2] on UT-Zappos. The results indicate that increasing any branch weight to 2 or decreasing it to 0.5 significantly reduces both HM and AUC, demonstrating that the model performance is relatively sensitive to the loss weights of the three branches. The detailed experimental results are as follows:
>
> | (βa, βo, βc) | Seen | Unseen | HM   | AUC  |
> |--------------|:----:|:------:|:----:|:----:|
> | (1, 1, 0.5)  | 66.2 | 74.4   | 55.6 | 42.9 |
> | (1, 1, 2)    | 66.5 | 74.5   | 55.7 | 43.0 |
> | (1, 0.5, 1)  | 67.2 | 73.6   | 55.0 | 42.0 |
> | (1, 2, 1)    | 65.2 | 73.2   | 54.0 | 40.0 |
> | (0.5, 1, 1)  | 66.2 | 73.6   | 54.4 | 41.3 |
> | (2, 1, 1)    | **70.0** | 75.3   | 57.6 | 44.5 |
> | (1, 1, 1)    | 67.6 | **76.1**   | **57.6** | **45.4** |
>
> As can be seen from the table, the (1,1,1) configuration achieves the best overall HM and AUC, while maintaining balanced performance across all three branches. This indicates that, in our model, keeping a balanced loss weighting among the attribute, object, and compositional branches is crucial, as excessively large or small weights can negatively affect performance. The results and discussions are added in Appendix Sec.G and Table 9. Thanks.

---

> > ### Author Response · Authors · 2025-11-23
> > **Point-to-Point Response to Reviewer th9S (2/2)**
> >
> > **Q4: Efficiency analysis.**
> >
> > **A4:** Sorry for this confusion. We provide the resource usage analysis below about UT-Zappos in Table 6. As shown in the table, compared with the lightweight baseline model, BAYECZSL exhibits an increase in both the number of parameters (14.9M) and memory usage (11.7G), primarily due to the construction of three-path prompt branches, which significantly enhance the model's expressive capacity. In terms of training efficiency, the incorporation of probability distribution learning, Monte-Carlo sampling, and other steps leads to a moderate increase in training time compared to the baseline, with a correspondingly slightly higher inference latency. Nevertheless, it is important to note that this additional overhead is manageable and does not substantially affect the overall inference efficiency.
> >
> > | Method                    | Params ↑ | Memory ↑ | Training time ↑ | Inference Speed ↑ | AUC ↑ |
> > |--------------------------|:--------:|:--------:|:----------------:|:------------------:|:-----:|
> > | Troika  | 21.7M   | 9.0G     | 15.1 min         | 22.0 ms            | 41.9  |
> > | Baseline                 | 8.7M     | 8.6G     | 11.8 min         | 14.2 ms            | 42.6  |
> > | **BayeCzsl (Ours)**      | **14.9M** | **11.7G** | **19.6 min**     | **25.1 ms**        | **45.4** |
> >
> > **Q5: Numerical stability in inverse-variance gaussian fusion.**
> >
> > **A5:** In our Compositional Distribution Synthesis, we employ inverse-variance weighted Gaussian fusion to compute the mean and variance of compositional prompts. To ensure numerical stability, we add a small regularization term (ε = 1e-6) to each posterior variance before computing its inverse. This prevents extremely small variances from causing numerical divergence or gradient explosion during training.
> >
> > Through our experiments, we observed that a moderate regularization has minimal impact on the fusion results while still preserving the correct relative weighting between attribute and object contributions. This strategy effectively balances numerical stability with the fidelity of semantic contributions, ensuring that the fused compositional distributions remain both reliable and expressive.
> >
> > ---
> >
> > We appreciate again your thoughtful review and we hope we addressed your concerns. Please let us know if you'd like any further information.

---

> ### Author Response · Authors · 2025-11-27
> **Looking forward to the discussion**
>
> Dear Reviewer,
>
> We sincerely appreciate the time and effort you have dedicated to reviewing our submission. We have submitted our rebuttal and would like to follow up to inquire whether our responses have sufficiently addressed your concerns.
>
> Please let us know if you have any remaining questions or require additional clarification. We value your feedback and are eager to ensure our work meets the highest standards.
>
> Authors

---

### Official Review · Reviewer_beGh · 2025-11-11

**Soundness:** 3
**Presentation:** 3
**Contribution:** 2
**Rating:** 4
**Confidence:** 4

**Summary:**

This paper proposes BAYECZSL, a Bayesian-induced framework for Compositional Zero-Shot Learning that models attribute and
object prompts as probability distributions rather than deterministic embeddings. The method captures intra-primitive diversity
and semantic uncertainty by learning Bayesian distributions over textual prompts, which are then fused through a Compositional
Distribution Synthesis module to form a compositional prompt space. A Three-path Distribution Enhancement module further
refines these distributions via invertible mappings, enabling more expressive sampling and richer semantic coverage.

**Strengths:**

1. The paper reformulates CZSL from a Bayesian inference standpoint, introducing the idea of learning probability distributions over primitive textual prompts. This probabilistic view allows the model to explicitly model intraprimitive variability and semantic uncertainty, addressing a key limitation of prior deterministic prompt-based methods.

2. The proposed CDS and TDE modules jointly enable a unified, expressive prompt distribution space. CDS fuses attribute and object distributions to model their semantic relationships, while TDE transforms base distributions into more flexible ones via invertible mappings.

**Weaknesses:**

1. Although the combination of Bayesian modeling and compositional synthesis is interesting, several prior works have explored distributional or probabilistic prompt spaces. E.g. "Prompt Distribution Learning" and "Prompting Language-Informed Distribution for Compositional Zero-Shot Learning". The contribution may thus be perceived as an evolutionary extension rather than a fundamentally new paradigm.

2. The baselines used for comparison are outdated, primarily consisting of works from 2022 and 2023. It would strengthen the paper to include evaluations against more recent state-of-the-art approaches.

3. Despite being compared with relatively outdated baselines, the reported improvements are not particularly significant.

**Questions:**

How many prompt tokens are used ?

---

> ### Author Response · Authors · 2025-11-23
> **Point-to-Point Response to Reviewer beGh (1/2)**
>
> Thank you so much for the valuable time and constructive feedback for us to improve the paper. We provide point-to-point response below.
>
> ---
>
> **Q1: Comparison with existing prompt distribution methods.**
>
> **A1:** First, thank you for acknowledging our interesting idea. However,we respectfully argue that BAYECZSL represents a principled Bayesian framework in Compositional Zero-Shot Learning (CZSL) that goes beyond prior prompt distribution-based work, rather than being an evolutionary extension. Below, we clarify three key distinctions between our work and existing prompt distribution-based approaches (*e.g.*, ProDA[ref1], PLID[ref2]):
>
> **1) Principled Bayesian framework for primitive distribution modeling**
>
> Prior prompt distribution-based works construct a diverse set of prompts or a prompt distribution using empirical methods or external knowledge from LLMs. Their motivation is primarily focused on prompt diversity. In contrast, our BAYECZSL is the first to model the prompts for the primitive concepts in CZSL as a Bayesian posterior distribution. We approach CZSL from the perspective of Bayesian inference, explicitly integrating the intrinsic uncertainty of the primitives into the model. This principled Variational Inference-based probabilistic modeling provides a strong theoretical foundation and regularization capability, which is crucial for tackling the generalization challenge of unseen compositions.
>
> **2) Compositional distribution modeling strategy**
>
> Prior prompt distribution-based works learns primitive representations independently and combining them using deterministic mechanisms or directly learn compositional distributions over the composed labels, ignoring the complex semantic relationship between the primitive distributions. In contrast, our BAYECZSL introduces the Compositional Distribution Synthesis (CDS) module, which aggregates the learned probability distribution of both attribute and object branches  to form the compositional prompt spac. This module allows us to explicitly capture the semantic relationships  and the propagation of uncertainty between the primitives.
>
> **3) Context-aware distribution learning**
>
> Prior prompt distribution-based works learn a static distribution for a concept (e.g., "old") that is applied universally across all images. They fail to explicitly account for context-dependency, where the meaning of "old" shifts significantly between "old dog" and "old town". In Contrast, our BAYECZSL introduces image-specific primitive distributions. Our model can dynamically adjust the prompt distributions based on the visual input. Thus our BAYECZSL generates more fine-grained and context-relevant prompt distributions to capture the primitive variation, which is a significant advancement over existing methods
>
> In conclusion, our contribution is not a mere evolutionary extension but a shift to **a principled Bayesian framework**, **a novel distribution-level composition strategy** and **context-aware distribution learning**, establishing a more robust paradigm for CZSL.
>
> [ref1] Prompt Distribution Learning. CVPR 2022.
>
> [ref2] Prompting Language-Informed Distribution for Compositional Zero-Shot Learning. ECCV 2024.
>
> **Q2: Incorporating more recent state-of-the-art approaches.**
>
> **A2:** Thanks for your suggestion. In our original submission, we had already compared our method against three representative approaches from 2024: GIPCOL [ref3], Troika [ref4], and PLID [ref5]. To make the comparison clearer, we additionally annotate each method in Table 1 and Table 2 with its publication venue and year.
>
> To ensure that our evaluation reflects the most recent progress in the field, we have incorporated two newly published state-of-the-art methods (ProLT [ref6] and CDS-CZSL [ref7], both from 2024) into our comparisons. We have updated it in Table 1 and 2. The experimental results show that our method maintains clear advantages over these stronger and more recent baselines.
>
> [ref3] Gipcol: Graph-injected soft prompting for compositional zero-shot learning. CVPR2024.
>
> [ref4] Troika: Multi-path cross-modal traction for compositional zero-shot learning. CVPR2024.
>
> [ref5] Prompting language-informed distribution for compositional zero-shot learning. ECCV2024.
>
> [ref6] Revealing the proximate long-tail distribution in compositional zero-shot learning. AAAI 2024.
>
> [ref7] Context-based and diversity-driven specificity in compositional zero-shot learning. CVPR 2024.

---

> > ### Author Response · Authors · 2025-11-23
> > **Point-to-Point Response to Reviewer beGh (2/2)**
> >
> > **Q3: On the performance gains of our method.**
> >
> > **A3:** Thank you for your comment. Under the Closed-World setting, BAYECZSL achieves improvements in AUC of **+0.4**, **+3.7**, and **+0.4** on MIT-States, UT-Zappos, and C-GQA, respectively. Overall, BAYECZSL exceeds the current state-of-the-art by up to **+1.8%**, **+8.9%**, and **+3.2%** relative AUC on these three benchmarks. Under the more challenging Open-World setting, BAYECZSL also demonstrates notable gains, improving AUC by **+0.3**, **+2.3**, and **+0.4** on MIT-States, UT-Zappos, and C-GQA, respectively. Importantly, BAYECZSL still surpasses the state-of-the-art method by up to **+4.1%**, **+7.0%**, and **+14.8%** relative AUC on the three datasets, demonstrating its strong robustness and overall competitiveness. These consistent improvements across multiple benchmarks and evaluation settings clearly demonstrate the robustness and superiority of our method over the state-of-the-art approaches.
> >
> > Across all datasets, BAYECZSL achieves stable and significant improvements over the baseline. On UT-Zappos, the final model improves HM by **+2.2** and AUC by **+2.8**. A similar trend is observed on MIT-States, where HM increases by **+2.3** and AUC by **+2.1**. These results also  demonstrate the robustness and superiority of our method.
> >
> >  In future work, we plan to incorporate our method into stronger baselines, such as CDS-CZSL [ref8], to further validate its effectiveness.
> >
> >  [ref8] Context-based and diversity-driven specificity in compositional zero-shot learning. CVPR 2024.
> >
> > **Q4: The number of prompt tokens.**
> >
> > **A4:** Sorry for this confusion. Similar to all CLIP-based prompt learning methods [ref9,ref10], the total number of tokens fed into the CLIP text encoder in our approach is 77. We replace only a subset of the token embeddings, specifically the context tokens, attribute tokens, and object tokens, with learnable embeddings following CoOp [ref9] and Trokia [ref10], while keeping the total length of the CLIP text sequence unchanged. We incorporate the above explanation into the implementation details in Section 4.1. Thanks.
> >
> > [ref9] Learning invariant visual representations for compositional zero-shot learning. ECCV 2022.
> >
> > [ref10] Troika: Multi-path cross-modal traction for compositional zero-shot learning. CVPR2024.
> >
> > ---
> >
> > We appreciate again your thoughtful review and we hope we addressed your concerns. Please let us know if you'd like any further information.

---

> ### Author Response · Authors · 2025-11-27
> **Looking forward to the discussion**
>
> Dear Reviewer,
>
> We sincerely appreciate the time and effort you have dedicated to reviewing our submission. We have submitted our rebuttal and would like to follow up to inquire whether our responses have sufficiently addressed your concerns.
>
> Please let us know if you have any remaining questions or require additional clarification. We value your feedback and are eager to ensure our work meets the highest standards.
>
> Authors

---

### Author Response · Authors · 2025-11-24
**Summary of Revisions**

We express our sincere gratitude to all reviewers for their valuable time and thorough assessment of our manuscript. We have revised our paper according to your comments (where the revised parts are highlighted in red color). The major changes are as follows:

1. We expand the comparison to include more recent state-of-the-art methods, according to Reviewer beGh's comments in Table 1 and 2.

2. We clarify the number and structure of prompt tokens used in our CLIP-based text encoder, according to Reviewer beGh's comments in Sec. 4.1.

3. We add experiments to evaluate the performance of variance-inverse weight Gaussian fusion strategy, according to Reviewer th9S's comments in Appendix Sec.I and Table 11, 12.

4. We add the extensive ablation module-wise experiments on MIT-States, according to Reviewer th9S's comments in Table 3(a).

5. We offer a detailed sensitivity analysis on the hyperparameter β, according to Reviewer th9S's comments in Appendix Sec.G and Table 9.

6. We add more details about the resource usage, according to Reviewer th9S's comments in Appendix Sec.H and Table 6.

7. we report results averaged over 5 random seeds, according to Reviewer QwoX's and Hjnp's comments in Tables 1, 2 and 3.

8. We add experiments to evaluate the performance of distribution enhancement strategy, according to Reviewer QwoX's comments in Appendix Sec. J and Table 14.

9. We add ablation study regarding composition posterior, according to Reviewer Hjnp's comments in Appendix Sec.I and Table 13.

10.  We add experiments to compare the performance between CoCoOp and our method, according to Reviewer Hjnp's comments in Appendix Sec.H and Table 10.

11.  We provide experiments to evaluate the performance on AAO-MIT-States, according to Reviewer Hjnp's comments in Appendix Sec.L and Table 16.

12.  We add experiments across different pre-trained backbones (ViT-B), according to Reviewer Hjnp's comments in Appendix Sec.K and Table 15.

13.  We update the description regarding performance improvement, according to Reviewer Hjnp's comments in Sec. 1.

14.  We clarify the mean of cross-branch synergies, according to Reviewer Hjnp's comments in Sec. 1.

15. We discuss the trade-off between the number of Monte Carlo samples and performance, according to Reviewer Hjnp's comments in Table 4 and Sec. 4.3.

Please refer to our response for more details. We have strived to address each of your concerns and welcome further discussions and insights.

Sincerely yours,

Authors.

---

### Author Response · Authors · 2025-12-04
**Summary of Reviews and Revisions**

We sincerely appreciate the Area Chair for their effort in handling the unexpected issues during the review process, and we are grateful to all reviewers for their valuable time and insightful feedback. In response, we have provided detailed point-by-point clarifications and revised the manuscript accordingly. We summarize the key aspects of our work, the reviewers’ comments, and our responses as follows.
***

**Summary of This Work**

This paper proposes BAYECZSL, a Bayesian-induced CZSL framework that learns probability distributions over each primitive textual prompt from a Bayesian perspective (Reviewer th9S). The method captures intra-primitive diversity and semantic uncertainty by learning Bayesian distributions over textual prompts, which are then fused through a Compositional Distribution Synthesis module to form a compositional prompt space (Reviewer beGh). Then, to model more complex distributions, it uses a three-path distribution enhancement module to transform the initial  distributions into flexible distributions using a sequence of invertible mappings. The authors conduct experiments on three standard CZSL benchmarks show state-of-the-art results in closed-world and open-world settings (Reviewer QwoX).
***

**Summary of Reviewers’ Comments and Our Responses**

**Reviewer beGh (Rating 4 & Confidence 4)**

Reviewer beGh notes that the paper "explicitly model intraprimitive variability and semantic uncertainty, addressing a key limitation of prior deterministic prompt-based methods." The main concerns relate to: i) comparison with existing prompt distribution methods (Q1); ii) incorporating more recent state-of-the-art approaches (Q2); iii) the performance gains of our method (Q3); and iv) the number of prompt tokens (Q4).

In response, we i) clarify three key distinctions between our work and existing prompt distribution-based approaches; ii) expand the comparison to include more recent state-of-the-art methods (Table 1 and 2); iii) clarify the reported performance gains; iv) clarify the number and structure of prompt tokens used in our CLIP-based text encoder (Sec. 4.1).

**Reviewer th9S (Rating 10 & Confidence 5)**

Reviewer th9S acknowledges the novelty and technical soundness of the methodology, and highlights the comprehensiveness of the experimental validation and the quality of the paper's presentation. The main concerns relate to: i) the choice of the inverse-variance weighted gaussian fusion strategy (Q1); ii) additional ablation experiments (Q2); iii) impact of the hyperparameter β (Q3); and iv) efficiency analysis (Q4).

In response, we i) add experiments to evaluate the variance-inverse weight Gaussian fusion strategy (Appendix Sec.I and Table 11, 12); ii) add the extensive ablation module-wise experiments on MIT-States (Table 3(a)); iii) offer a detailed sensitivity analysis on β (Appendix Sec.G and Table 9); iv) add more details about the resource usage (Appendix Sec.H and Table 6).

**Reviewer QwoX (Rating 6 & Confidence 4)**

Reviewer QwoX regards this probabilistic framing as both clear and technically sound, underpinned by a principled probabilistic formulation and supported by solid empirical evidence. The main concerns relate to: i) experimental reproducibility (Q1); ii) diagonal-gaussian residual assumption and distribution enhancement (Q2).

In response, we i) report results averaged over 5 random seeds (Tables 1, 2 and 3); ii) clarify the assumption ensures computational efficiency and training stability and add experiments to evaluate the distribution enhancement strategy (Appendix Sec. J and Table 14).

**Reviewer Hjnp (Rating 4 & Confidence 4)**

Reviewer Hjnp acknowledges the high clarity and strong motivation of the work, and highlights the method's superior empirical performance over prior work. The main concerns relate to: i) novelty of our BAYECZSL (Q1); ii) ablation study regarding composition posterior (Q2); iii) comparison with CoCoOp (Q3); iv) more evaluation on AAO-MIT-States (Q4); v) evaluation results across different pre-trained models (Q5).

In response, we i) clarify the key distinctions between our work and existing approaches in terms of the motivation and technical design; ii) add ablation study regarding composition posterior (Appendix Sec.I and Table 13); iii) add experiments to compare the performance between CoCoOp and our method (Appendix Sec.H and Table 10); iv) provide experiments on AAO-MIT-States (Appendix Sec.L and Table 16); v) add experiments across different pre-trained backbones (ViT-B) (Appendix Sec.K and Table 15).

Following our response, Reviewer Hjnp states that the supplementary experiments and clarifications fully address their concerns and raises their score accordingly.
***

Detailed explanations are provided in the point-by-point responses, and all corresponding revisions are highlighted in red throughout the revised manuscript. Thank you again for your time and consideration.

---

### Meta-Review · Area_Chair_RNCJ · 2025-12-24

**Summary:**

This paper proposes a prompt learning method for Compositional Zero Shot Learning (CZSL). Its motivation stems from the existence of a distribution offset for an attribute or object across different categories. To address this, it employs Bayesian modeling to capture image-specific distribution offset and correct the text prompts accordingly. The paper received initial scores of 10, 6, 4, and 4.

The two negative-score reviewers express concerns about insufficient innovation, noting that the core methodology proposed is similar to the workflow in [1]. Additionally, the methods compared in the experimental results are from before CVPR 2024. A significant number of recent publications are omitted, making it hard and seemingly unfair to assess the paper's advancement. The two reviewers who gave positive feedback request additional experiments, which, while improving the paper's quality, do not address the aforementioned critical issues. Meanwhile, the score 10 reviewer provided only seemingly general comments, lacking sufficient detail to support the 10-point evaluation and the 5-point confidence level. I was not convinced.

Given the above summary, I do not think this paper meets the acceptance bar of ICLR.

[1] Prompt Distribution Learning. CVPR 2022.

**Reviewer Concerns:**

A. Points that are partially addressed/mitigated in the rebuttal.
1. Additional comparative experiments and ablation studies: The authors expand their comparative experiments to include more datasets and conduct ablation studies on other technical details.
2. Additional parameter and cost analysis: The authors incorporate further sensitivity analysis of more hyper-parameters and computational cost analysis.

B. Remaining concerns
1. Similar core workflow: Multiple reviewers point that the workflow of the methods in this paper, i.e., Bayesian modeling + Monte Carlo sampling, is similar to [1]. The authors emphasize differences in motivation and technical details, but it remains unclear whether these constitute major innovations.
2. Outdated comparison methods: The methods used in the experiments are outdated. And the authors fail to incorporate the latest methods in their response. Consequently, the timeliness of this paper is challenged.

**Reviewer Scores:**

For Reviewers th9S (Rating: 10) and QwoX (Rating: 6), their primary concerns are the inadequacy of the ablation experiments. The authors have supplemented additional ablation experiments in their response. I think this may lead them to maintain or increase their scores.

For Reviewers beGh (Rating: 4) and Hjnp (Rating: 4), their primary concerns are the lack of methodological innovation and the outdated nature of the comparison method. The authors reiterate the strengths of the proposed method in their response. However, given that the core aspects of its technical paradigm and methodology are not novel, the reviewers may not increase their scores. Furthermore, the authors failed to provide an effective response regarding the outdated comparison method.

---

### Decision · Program_Chairs · 2026-01-26

Reject